# Towards Foundation Models
# for Scientific Machine Learning:
# Characterizing Scaling and Transfer Behavior

**Shashank Subramanian**
Lawrence Berkeley National Lab
shashanksubramanian@lbl.gov

**Peter Harrington**
Lawrence Berkeley National Lab
pharrington@lbl.gov

**Kurt Keutzer**
UC Berkeley
keutzer@eecs.berkeley.edu

**Wahid Bhimji**
Lawrence Berkeley National Lab
wbhimji@lbl.gov

**Dmitriy Morozov**
Lawrence Berkeley National Lab
dmorozov@lbl.gov

**Michael W. Mahoney**
ICSI, LBNL, and Department of Statistics
mmahoney@stat.berkeley.edu

**Amir Gholami**
ICSI, UC Berkeley
amirgh@berkeley.edu

## Abstract

Pre-trained machine learning (ML) models have shown great performance for a wide range of applications, in particular in natural language processing (NLP) and computer vision (CV). Here, we study how pre-training could be used for scientific machine learning (SciML) applications, specifically in the context of transfer learning. We study the transfer behavior of these models as (i) the pre-trained model size is scaled, (ii) the downstream training dataset size is scaled, (iii) the physics parameters are systematically pushed out of distribution, and (iv) how a single model pre-trained on a mixture of different physics problems can be adapted to various downstream applications. We find that—when fine-tuned appropriately—transfer learning can help reach desired accuracy levels with orders of magnitude fewer downstream examples (across different tasks that can even be out-of-distribution) than training from scratch, with consistent behaviour across a wide range of downstream examples. We also find that fine-tuning these models yields more performance gains as model size increases, compared to training from scratch on new downstream tasks. These results hold for a broad range of PDE learning tasks. All in all, our results demonstrate the potential of the "pre-train and fine-tune" paradigm for SciML problems, demonstrating a path towards building SciML foundation models. Our code is available as open-source at [1].

## 1   Introduction

Foundation models have received considerable interest recently [3]. This terminology refers to certain models that are trained on extremely large and diverse quantities of data and applied to a wide range of tasks. Rather than being designed for any single task, a foundation model serves as a "prior"

37th Conference on Neural Information Processing Systems (NeurIPS 2023).

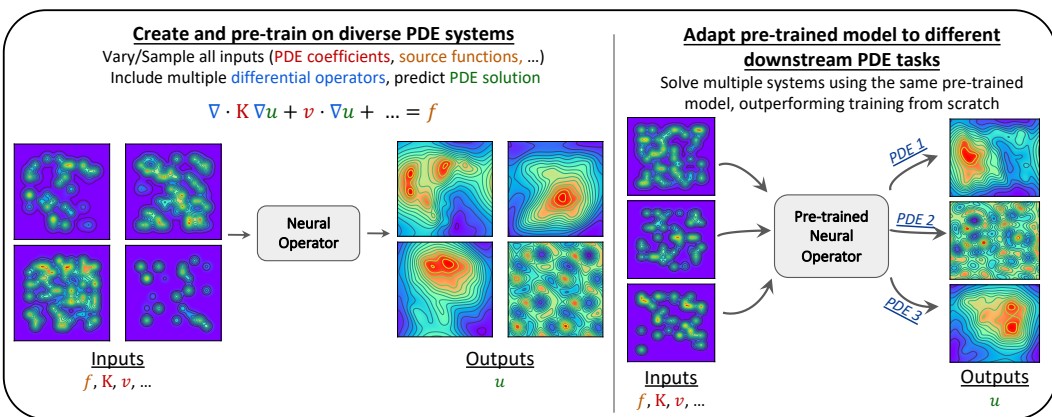

**Figure 1:** *Our setup consists of creating diverse training datasets, sampling both PDE coefficients and source functions simultaneously with different PDE operators and input data (coefficients, sources) distributions for pre-training. A neural operator is then pre-trained to predict the PDE solutions given these inputs and the ground truth solutions (computed through PDE solvers). The pre-trained model is then adapted with minimal fine-tuning (zero-shot or few-shot), and it is used in various downstream tasks (PDE systems) that can be in-domain or out-of-domain from the pre-training datasets. The pre-training with multiple solution operators allows the same model to transfer to several very different systems. For instance, PDE 2 (Helmholtz) manifests highly oscillatory solutions compared to PDE 1 (Advection-Diffusion) or PDE 3 (Poisson's). We further characterize the scaling and transfer properties of this model as a function of downstream data scale and model size scale.*

or "foundation" upon which other models can be built. It does so by using transfer learning (TL) methods to fine-tune or adapt the foundation model to a wide range of downstream tasks, using minimal additional data for each additional task. Perhaps the most well-known foundation models are pre-trained large-language models (LLMs) such as BERT [11] and the GPT models [38, 39, 5]. The scaling with respect to the amount of data, the size of the model, and the amount of compute [23, 20, 21] is key to the training of these models. An important aspect of a trained foundation model is the ability to perform tasks seemingly different than those for which it was trained by leveraging shared features across the training tasks. This approach to model development is quite different than the traditional approach of training a one-off model from scratch for each specific problem and each specific dataset. Naturally, it is of interest how broadly this methodological approach can be applied.

Scientific machine learning (SciML) [45] is an area that combines tools from ML and scientific computing to address domain-specific scientific and engineering challenges. It holds promise to drive the next wave of data-driven discovery in the physical and engineering sciences. Recent work has highlighted the promise [42, 32, 30, 24] as well as some of the many challenges [28, 12] of developing SciML models—in general as well as with the traditional one-off learning approach. Many SciML models emulate physical systems described by Partial Differential Equations (PDEs). For example, Physics-Informed Neural Networks [42, 28] impose the PDE as a soft penalty in the loss function. However, they are restricted to solving a single instance of the PDE. The Neural Network (NN) needs to be retrained for each new set of PDE physics coefficients, sources, and/or initial/boundary conditions (IC/BCs). Subsequent models have been developed to learn the full solution operator [30, 32] by training across different coefficients (and/or initial and boundary conditions). These *neural operators* learn mappings between two function spaces from a finite collection of input-output pairs (that represent the coefficients/initial or boundary conditions as the input and the PDE solution function as the output). This makes them more general and versatile in emulating any PDE system. However, with new coefficients/sources or new differential operators, they too need to be retrained from scratch.

In this paper, we adopt and evaluate the methodology that has been applied successfully in CV and NLP to develop foundation models, with the goal of determining whether such a model is even possible for SciML problems. In particular, we provide an extensive analysis of the scaling and TL behavior of neural operators trained on diverse training datasets from multiple PDE systems. An important aspect of this approach is to explore several dimensions that include the model (architecture and scale), data (diversity and scale), training recipes (pre-training and fine-tuning), and out-of-

distribution (OOD) transfer behaviour. For transfer behavior, a major difference from CV/NLP is the diversity of scales, features, and solution behaviors across PDEs. For example, while OOD shifts in CV/NLP typically involve things like different color distributions or semantic contexts, in SciML, it is possible to get significantly different behavior of a PDE solution as physics parameters and conditions change that manifest as novel features (for example, increased emphasis on very different frequency scales of the solution causing significant shifts in detail, sharpness, and other image attributes). For LLMs, given the maturity of the NLP community within ML, these dimensions are well-explored. For SciML problems, in contrast, all these dimensions are open questions. Here, we explore several of these questions. We do so in the context of a specific model architecture, namely, the *Fourier Neural Operator (FNO)*, as a prototypical SciML model that has demonstrated promising results modeling PDEs across a wide range of scientific applications [30, 37, 49, 53, 29]. We focus on the scaling and TL behavior of the FNO on common PDE systems that include Poisson's, Advection-Diffusion, and Helmholtz PDE systems. These systems underpin a wide range of physical systems: fluid flow systems; biological simulations; wave propagation systems; and many others.

See Fig. 1 for a schematic summary of our methodological approach. Our main results demonstrate the potential of the "pre-train and fine-tune" paradigm for SciML problems, demonstrating a path towards building SciML foundation models. In more detail, our main contributions are the following.

1. **Pre-training dataset generation.** We develop a large suite of datasets, and we train our models on data where all the variables (inputs) of any PDE operator are sampled. This is an important step towards developing NNs that can generalize across a variety of downstream tasks, and it extends several previous works, including the original FNO [30], where certain inputs are kept fixed. Not sampling can trivially push the neural operator OOD if, e.g., the source function was changed. We study transfer learning to both in-domain and out-of-domain distributions, characterized by different samplings of PDE coefficients (diffusion, advection, wavenumbers, etc.) and inhomogeneous source functions. We emphasize the construction of pre-trained datasets with sufficient diversity as well as normalization strategies, without which we observe significant degradation of performance.

2. **Downstream (target) data scaling.** We study the effect of the scale of downstream (target) data in TL performance from the pre-trained model. Here, we assume that a large amount of data is available for pre-training, and data are limited for the downstream tasks (as in many scientific examples); and we are interested in reaching desired accuracy levels with the least amount of additional downstream data. We consider both zero-shot and few-shot TL: zero-shot is the direct evaluation of the pre-trained model on the downstream dataset; and few-shot involves using O(10) downstream data to fine-tune the pre-trained model. While few-shot typically refers to prompting with in-context examples during inference in NLP, here we define it as fine-tuning the model with a few examples. We observe that TL from the pre-trained model can lead to significant performance gains over training the model from scratch on the downstream (target) data, with orders of magnitude less data needed to reach a desired accuracy level (see Fig. 2). We observe this gain over a wide range of data scales, until we enter the "large target data" regime (as much data as pre-training), where we observe similar accuracies for TL as training from scratch.

3. **Model (parameter) size scaling.** We study the parameter scaling of the model by scaling our model sizes from 64K to 256M parameters (a multiplicative factor of 4K). We observe an error saturation at small model sizes (due to insufficient model expressivity) that monotonically drops as we increase the model size. While both fine-tuned models and models trained from scratch exhibit gains with increased model size, we observe that fine-tuning achieves greater performance gains with parameter scaling (see Fig. 3).

4. **Transfer learning behavior over underlying physics.** We study the effect of varying the underlying physics in the target domain. In SciML (unlike traditional non-scientific ML), there are typically fundamental constraints such as conservation laws that govern the behaviour of the solution. In some cases, we may even have access to parameters of the underlying physical constraints, and thus a physical understanding of the distribution of the data. It is natural, and necessary, to systematically quantify the effect of these parameters as our downstream tasks go OOD, as this provides a good way to test the (OOD) generalization capability of pre-trained models for SciML applications. We find that for in-distribution TL, the pre-trained model can significantly outperform a model trained from scratch, irrespective of how many new data examples were used for fine-tuning, until the large target data regime (e.g., see Fig. 4a), showing orders of magnitude better accuracies than training from scratch. We also observe these gains for downstream tasks that are moderately OOD, with few-shot fine-tuning providing again orders of magnitude better

accuracies (see Fig. 4b, Fig. 4c). As we systematically go further OOD (see the quantification in Tab. 1), we observe the performance gains expectedly reduce, with more significant drop in the low data regimes (e.g., see Fig. 4d).

5. **Transfer learning behavior over multiple operators.** We study the effect of simultaneous pre-training on multiple PDE systems that exhibit qualitatively different solution behaviors (e.g., Poisson's and Helmholtz operator solutions show very dissimilar patterns, given a source function). We include the coefficient/source functions for all the operators as inputs to the model, with zero values if those terms do not exist in a given PDE instance. During inference, the zero inputs restrict the neural network to make predictions for the correct operator (see §4 for details). Among other things, we show that the same model pre-trained on different operators retains its gains across different downstream tasks (see Fig. 5), paving the way for it to be used in the foundational sense.

## 2 Related work

In recent years, there has been widespread interest in modeling PDE systems with neural operators or operator learning in a broad variety of science and engineering applications [32, 30, 27, 41, 33, 37, 25, 2, 49, 4, 29, 26]. Following the success of TL in CV and NLP tasks [54, 43, 48, 13, 36, 40, 22], there have been several investigations into how TL can be leveraged for SciML problems involving differential equations. Most have focused on applications of Physics-Informed Neural Networks (PINNs) [35, 19, 15, 8, 18, 17, 50, 14, 10, 31], where models can be fine-tuned or adapted using a physics-based loss function determined by the specific target PDE/ODE system. Another common theme is evaluating how well TL can account for the diversity of geometries [6, 47, 50, 16] and discretizations [34, 7, 44, 46] found in scientific computing. Recently, attention has also been devoted to TL for neural operators in SciML, where some works have explored certain components of our analysis in isolation. This includes studies on the TL performance of DeepONet [51, 16, 55] and FNO [31, 9], where one of either the target domain, PDE coefficients, or PDE source functions shift, are varied. In [9], the authors evaluate different model and training dataset sizes for different neural operator architectures, but their analysis is mostly limited to within-distribution experiments and single operators, and they do not analyze the fine-tuning performance of such models (particularly on OOD data). The works above have not considered the pre-training regimes over a diverse set of physical systems and most have not characterized their TL as a function of data and model scaling, nor included multiple PDE operators during training. To the best of our knowledge, the work of [52] (which has appeared concurrent to this work) is the first (along with ours) to consider the TL potential of neural networks across operators and data distributions. There, the authors adapt in-context learning (from LLMs) to solve differential equations with transformer-based models. The main difference with [52] is that the authors focus on in-context learning which requires prompting (with up to five example demos) to solve OOD tasks including different differential equation coefficients and operators than those seen at train time (similar to our work). In contrast, we focus on TL with zero-shot and few-shot learning through fine-tuning. Thus the two approaches (TL through fine tuning vs in-context learning) are complementary but different. We also note that their investigation was performed at a much smaller scale (from both model and data perspective), and was limited to simpler dynamical systems, as compared to the scales tested in our work. Overall, the prior related work provide important initial results on the interpolation and extrapolation behavior of neural operators in the context of differential equations, and/or limited investigation into the behavior as model and dataset sizes are increased. However, none of them consider these aspects simultaneously with a more diverse pre-training corpus (by varying all input variables such as source functions and PDE coefficients and/or including different operators for pre-training), which is closer to methodology adopted by CV and NLP in the development of their foundation models, with emphasis on the importance of characterizing scaling properties [23, 21].

## 3 Methods

Pre-training a foundation model requires that we first collect a large amount of diverse training data and then train a base model that could subsequently be used with TL for a downstream application. There are several possible methods for TL of the foundation model. One approach is in-context learning, where the foundation model is prompted with few-shot input-output examples for the downstream problem followed by the target input. The model then sees these examples and learns how to compute the target output. This is the approach used by modern NLP models such as

GPT [38, 39, 5] as well as the work of [52]. While this approach is very useful for cases with very few training datapoints available, it is often better to fine-tune a foundation model for a downstream task, when one has access to more downstream training data. This is also supported by NLP models and it often results in better performance—if enough training data is available. We focus on the latter setup, and we study how the TL performance behaves for different problem setups. Our goal is to understand the different moving parts associated with training a foundation model for SciML applications and, specifically, the impact of model scaling, dataset size, and different physics involved in the problem. Below, we discuss (i) the different physics operators considered and our pre-training setup that includes the training dataset generation, (ii) NN model architecture setup for training and inference, and (iii) performance metrics.

**PDE/physics system setup.** We consider three PDE systems that are common building blocks for many scientific application: 2D Poisson's; Advection-Diffusion; and Helmholtz equations. These PDE systems can be formulated as follows:

1. *Poisson's* (SYS-1): We consider a prototypical elliptic system with periodic boundary conditions in domain $\Omega = [0, 1]^2$:
$$-\text{div}\,\boldsymbol{K}\nabla u = f \quad \text{in } \Omega, \tag{1}$$
where $u(\boldsymbol{x})$ is the solution (state) function, $f(\boldsymbol{x})$ is a source (forcing) function, and $\boldsymbol{K}$ is the diffusion coefficient tensor. We use $\boldsymbol{K}$ to quantify the physics of this system.

2. *Advection-Diffusion* (SYS-2): We also consider a steady-state advection-diffusion equation that illustrates competing physical processes (advective and diffusive processes) through two differential operators. We use periodic boundary conditions in domain $\Omega = [0, 1]^2$:
$$-\text{div}\,\boldsymbol{K}\nabla u + \boldsymbol{v}\cdot\nabla u = f \quad \text{in } \Omega, \tag{2}$$
where $u(\boldsymbol{x})$ is the solution (state) function, $f(\boldsymbol{x})$ is a source (forcing) function, $\boldsymbol{K}$ is the diffusion coefficient tensor, and $\boldsymbol{v}$ is the velocity vector. To quantify the competing advective/diffusive scales of this system, we define the ratio of advection to diffusion as $\Psi = \|\boldsymbol{v}\cdot\nabla u\|/\|\text{div}\,\boldsymbol{K}\nabla u\|$.

3. *Helmholtz* (SYS-3): Finally, we also consider the inhomogeneous Helmholtz equation with periodic boundary conditions in domain $\Omega = [0, 1]^2$. We take this as an example challenging system that can exhibit high-frequency oscillatory spatial patterns that can be difficult to generalize. This system is formulated as:
$$-\Delta u + \omega u = f \quad \text{in } \Omega, \tag{3}$$
where $u(\boldsymbol{x})$ is the solution (state) function, $f(\boldsymbol{x})$ is a source (forcing) function, $\omega > 0$ is the wavenumber used to quantify the underlying physics of this system.

**Data setup.** For the above PDE systems, we are interested in (i) a large and diverse training dataset for pre-training and (ii) several downstream datasets (tasks) to quantify the TL performance. Given that we can solve these PDEs numerically, we can generate diverse set of training and testing datasets in a controllable fashion by varying the different parameters in these PDEs. In particular, we vary the following parameters for dataset generation:

(i) *Source function sampling:* We sample different source functions $f \sim \mathcal{S}(\sigma, s)$, where $\mathcal{S}$ is a distribution that generates diverse and heterogeneous functions. (see Fig. 1 and Fig. A.1 for examples). Here, $\mathcal{S}$ represents a parameterization of the source function as a linear combination of $n_g$ radial (Gaussian) basis functions $\{\phi_i(\boldsymbol{x})\}_{i=1}^{n_g}$, where $\phi_i(\boldsymbol{x}) = \phi(\boldsymbol{x} - \boldsymbol{x_i})$ is a Gaussian function centered at grid point $\boldsymbol{x}_i$. Specifically: $f(\boldsymbol{x}) = \sum_{i=1}^{n_g} \phi_i(\boldsymbol{x})p_i$, with $\boldsymbol{p} = \{p_i\}_{i=1}^{n_g}$ as the parameterization vector. The spatial profile controlled by $\sigma$, the standard deviation of the Gaussian function, is preset to a small value to encourage high variability. Examples are sampled by uniformly randomly sampling $p_i \sim \mathcal{U}(0, 1)$. We further introduce heterogeneity by controlling the sparsity $s$ of $\boldsymbol{p}$ ($s$ defined as the number of non-zero components; see Appendix §A.1 for full details).

(i) *PDE coefficient sampling:* In SYS-1, we sample diffusion coefficient tensors $\boldsymbol{K} \sim \mathcal{K}(\lambda)$, where $\mathcal{K}$ is a distribution that generates varying scales of anisotropy and spread in the diffusion process: $\boldsymbol{K} = \boldsymbol{R}^{-1}\boldsymbol{D}\boldsymbol{R}$ with $\boldsymbol{D} = \text{diag}(1, e)$ and $\boldsymbol{R} = \text{rot}(\theta)$, where $e$ is an eigenvalue of the tensor that controls the anisotropy and extent of diffusion and $\text{rot}(\theta)$ is a rotation matrix with angle $\theta \sim \mathcal{U}(0, 2\pi)$ that controls the general diffusion direction. In SYS-2, we additionally also sample the velocity vector $\boldsymbol{v}$ direction from $\mathcal{U}(0, 2\pi)$. The ratio of advection to diffusion $\Psi$ is changed by scaling the velocity. In SYS-3, we sample the wavenumber $\omega$ as uniform

integers. We visualize the sampling process and resulting solutions with different ranges of underlying physics in Fig. A.1.

For each of the problems that we consider in the results section, we generate $2^{15}$ input-output samples (pairs) of data, where the inputs include the source $f$ as well as any PDE coefficients $(\boldsymbol{K}, \boldsymbol{v}, \omega)$, along with $2^{12}$ validation and testing samples each. The validation dataset is used for hyperparameter optimization, and the testing dataset is used for quantifying model performance. The pre-trained model is trained on the $2^{15}$ training examples.

We then perform different experiments to evaluate how this model can adapt/TL to different downstream tasks whose data could come from the following distributions: (i) same distribution as in the pre-training dataset (i.e., different input/output pairs but drawn from the same distribution of PDE coefficients/sources); and (ii) the harder task of adapting/TL to a downstream problem that can have slight/large deviation from the dataset used to pre-train the model. For the latter, we create the OOD data by keeping the PDE operator the same as the pre-training task, but sample the coefficients from a different range as in the pre-training. Given this dataset, we then study the TL behaviour for each case (both within distribution and OOD) by scaling both the downstream dataset size, as well the model architecture size which is discussed next.

**Pre-training method for training and inference.** The inputs to our model are 2D spatial functions discretized at $h \times w$ and represent the sources and PDE coefficients.[1] These input discretized functions are batched together to form an input tensor in $\mathbb{R}^{h \times w \times c}$. The output of the model is the numerical solution of the PDE in $\mathbb{R}^{h \times w}$. For the model architecture, we consider the FNO (details in Appendix §A.2). This model bears similarities to both vision transformer-like architectures (fixed image/feature resolution across the depth of the network) and convolutional architectures (successive global convolutions facilitated via FFTs). FNO is also a good choice for our setup as the problems we consider all have periodic boundary conditions (the FNO can also be adapted for non-periodic boundaries). The main modification that we make to the FNO is to incorporate a per-instance normalization layer in the model. We found that this is a critical component as the norm of the input data spans a wide range of values (up $100\times$ for our dataset). Please see Appendix §A.5 for details. Furthermore, we consider an $\ell_2$ loss function for pre-training. That is, the model is trained to predict the output solution given the input by minimizing a mean-squared error loss between the prediction and the ground truth.

To test how this pre-trained model can adapt to different downstream applications we consider the following cases. First, we consider zero-shot adaptation where the pre-trained model is tested on a downstream application without any fine-tuning. Second, we consider few-shot fine-tuning, where the pre-trained model can be fine-tuned on the downstream training dataset. We consider varying sizes for this downstream dataset size. Ideally, we prefer the pre-trained model to achieve good performance with as few examples as needed from the downstream application. We perform an ablation study by training a model from scratch on the downstream dataset alone to see how much gain can be achieved by using a pre-trained model. We also test how the pre-training adaptation performance changes as we scale the model size. This is motivated by observations in NLP [23] where we expect larger models to show better adaptation performance. To do the model scaling, we focus on two model hyperparameters: the embedding dimension $d$ and the number of Fourier modes used in the FNO $m$. (See Appendix §A.2 for the details.) We first fix $d = 16$ and scale $m \in \{4, 16\}$, then fix $m = 32$ and scale $d \in \{32, 128\}$ to approximately increase the parameter count $16\times$ in each scaling experiment from $64K$ to $256M$ parameters.

The main limitation of our work is that we focus on only the FNO model, whereas several other architectures (e.g. ViT models, DeepONets) exist. As discussed in §1, the FNO has been applied to a wide range of applications in SciML and is representative of a performant SciML architecture– restricting our analysis to one model architecture makes the scope of this study more feasible. However, this analysis needs to be done across other models architectures, as well.

## 4 Results

Our main results are the following. We demonstrate that pre-training a model on a diverse corpus of data and then fine-tuning it on downstream tasks leads to significantly better performance than training

---

[1]Constant coefficients are simply replicated across the $h \times w$ dimensions.

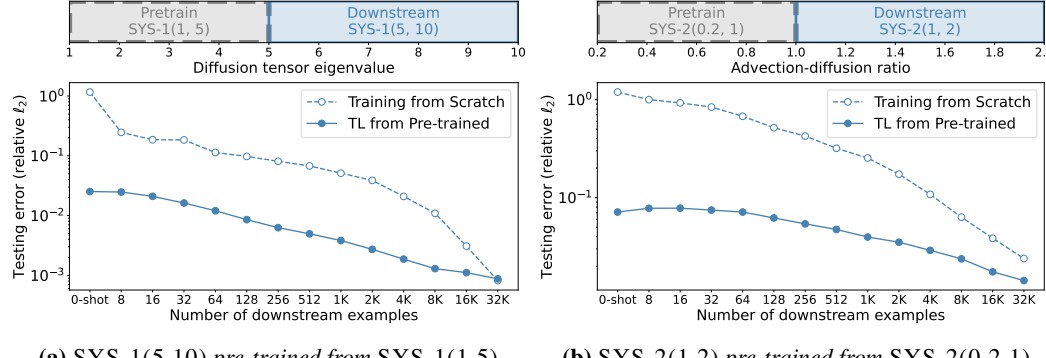

**(a)** SYS-1(5,10) *pre-trained from* SYS-1(1,5)   **(b)** SYS-2(1,2) *pre-trained from* SYS-2(0.2,1)

**Figure 2:** *Addressing (Q1). Testing error as a function of downstream examples for* SYS-1 *and* SYS-2. *We visualize the distribution of pre-training and downstream dataset physics at the top to illustrate (and quantifiy) the extent of distributional shifts. We observe excellent zero-shot and few-shot TL performance of the pre-trained model despite the modest OOD shifts and in medium-data regimes about $100\times$ increase in data efficiency. We observe diminishing returns from pre-training at the large-data regime ($O(2^{15})$ examples), which has as many examples as used in pre-training.*

a model from scratch. This holds even when the downstream data falls outside of the pre-training distribution, including when different physics models are combined. The advantage of pre-training is especially pronounced when the downstream data is limited, which is the most significant setting in practice, motivating the creation of foundation models for scientific machine learning.

To justify these conclusions, we focus on four key questions. What is the effect of **(Q1)** downstream dataset size and **(Q2)** neural operator model parameter size on TL? What is the TL behavior of the neural operator **(Q3)** over the underlying physics and **(Q4)** over multiple solution operators?

**(Q1): Downstream dataset scaling.** For SYS-1, we consider the pre-training system SYS-1(1,5), with diffusion constructed by sampling eigenvalue $e \sim \mathcal{U}(1,5)$ of the diffusion tensor $\boldsymbol{K}$. See Fig. A.1 for visualizations. This represents near isotropic to $5\times$ anisotropic diffusion. We use $e \sim \mathcal{U}(5,10)$ as the downstream dataset SYS-1(5,10). This represents $5\times$–$10\times$ anisotropic diffusion and is moderately out-of-distribution (OOD) from the pre-training dataset. While we systematically quantify the OOD effects in **(Q2)**, we use this specific test-case to illustrate the effect of the size of the downstream dataset. We plot the behaviour of testing error as a function of downstream examples in Fig. 2a. We train an FNO model for each number of downstream examples (x-axis on the plot) starting from "scratch" (random initialization) as well as from the pre-trained model parameters, with tuned hyperparameters for each experiment (see details in Appendix §A.4).

We illustrate the extent of distributional shift between the pre-training and downstream datasets through the range of diffusion tensor eigenvalue $e$—in this test case, a modest shift with no overlap (but relatively close). The testing error monotonically decreases as more downstream data is used for training, as we expect. The zero-shot TL shows excellent performance despite the moderate OOD shift of downstream examples. When training from scratch, we define "zero-shot" predictions as the output of the model with random initialization. With "few-shot" learning ($O(10)$ downstream examples), we observe a consistent performance increase over training from scratch. Given a desired error for downstream performance, TL from the pre-trained model can require orders of magnitude less data—for example, a desired error of 1E-2 needs only about 64 downstream data examples for fine-tuning, whereas training from scratch requires $8K$ (about $100\times$ more) examples to reach the same accuracy level. We further explore this in Appendix §B.1 and find the pre-trained model generally saves $O(1K - 10K)$ data compared to training from scratch in the few-shot learning setting, and outperforms training from scratch at all scales. Finally, with greater amounts of downstream data, the pre-training provides consistent performance gains until we enter the "large target data" regime, where the number of fine-tuning examples approaches the size of the entire pre-training dataset, and we see diminishing returns from pre-training. We repeat this experiment for SYS-2, using the following test-case: the pre-training system SYS-2(0.2,1) consists of advection-to-diffusion rates $\Psi \in \sim(0.2,1)$, representing about $1\times$–$5\times$ diffusion (relative to advection). For the downstream test, we use a modest TL with SYS-2(1,2) with $\Psi \in \sim(1,2)$ representing $1\times$–$2\times$ advection

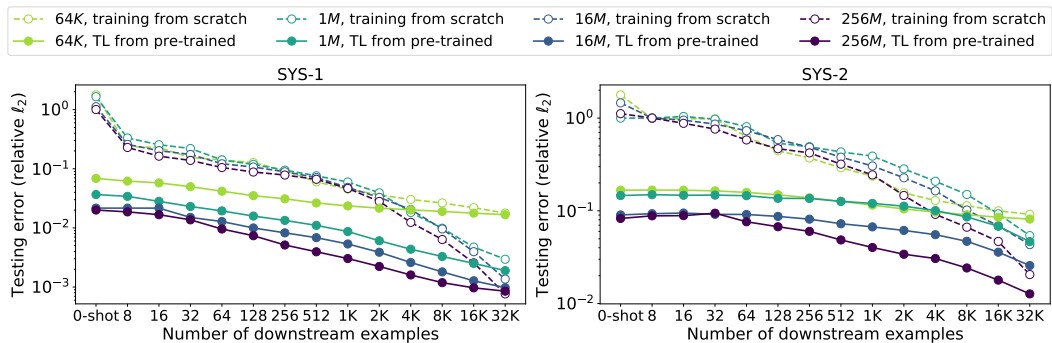

**Figure 3:** *Addressing (Q2). Model size scaling for* SYS-1 *and* SYS-2 *from* $64K$ *to* $256M$ *parameters for medium OOD test-cases. While finetuning consistently improves the model performance and data efficiency, we observe higher errors for small parameter regimes at* $64K$ *due to insufficient model capacity. The performance gains are significantly boosted through finetuning with a larger model set sizes monotonically up to* $256M$ *parameters.*

(relative to diffusion). We visualize this shift in Fig. 2b (top) and also show the testing error as a function of downstream examples. Both experiments reveal the same trend: TL delivers much higher performance, which improves with fine-tuning up to a point of diminishing returns.

**(Q2): Model (parameter count) scaling.** As described in our model setup, we vary the embedding $d$ and maximum Fourier modes $m$ to approximately increase the parameter count $16\times$ in each scaling experiment from $64K$ to $256M$ parameters. For models trained from scratch, we repeat the data scaling experiments for each parameter count. For the pre-trained model, we first identify the ideal hyperparameters (through grid-search hyperparameter tuning) for each model scale and repeat the above training experiments. We visualize the testing errors as a function of downstream examples used for SYS-1(5,10) (pre-training dataset used: SYS-1(1,5) signifying moderate OOD) for the different model scales in Fig. 3 (left). At the $64K$ parameter regime, the model capacity is insufficient, with large errors (greater than 1E-2) for either training recipe across the whole range of downstream example counts. As we move to larger models, both training from scratch and fine-tuning show higher performance that increase with more examples. Fine-tuning the pre-trained model boosts its performance, compared to training from scratch, as we increase the model scale and particularly across a wide range of downstream example counts (with $256M$ parameter model showing the least errors). We repeat the model scaling process for Advection-Diffusion SYS-2 (pre-training dataset SYS-2(0.2,1) with moderately OOD SYS-2(1,2) for downstream) and observe similar trends in Fig. 3 (right).

**Table 1:** *Different downstream datasets and extents of overlap with the pre-training dataset for* SYS-1 *and* SYS-2*, controlled by extent of anistropy (eigenvalue e) in diffusion tensor for* SYS-1 *and amount of advection relative to diffusion (ratio* $\bar{\Psi}$*) for* SYS-2.

| Pre-training | Downstream | Shift |
|---|---|---|
| SYS-1(1,5): $e \sim \mathcal{U}(1,5)$ | SYS-1(1,2.5): $e \sim \mathcal{U}(1, 2.5)$ | None |
| | SYS-1(2.5,7.5): $e \sim \mathcal{U}(2.5, 7.5)$ | Mild |
| | SYS-1(5,10): $e \sim \mathcal{U}(5, 10)$ | Med |
| | SYS-1(10,20): $e \sim \mathcal{U}(10, 20)$ | Large |
| SYS-2(0.2,1): $\Psi \in \sim(0.2, 1)$ | SYS-2(0.2,0.4): $\Psi \in \sim(0.2, 0.4)$ | None |
| | SYS-2(0.2,0.4): $\Psi \in \sim(0.2, 0.4)$ | Mild |
| | SYS-2(1,2): $\Psi \in \sim(1, 2)$ | Med |
| | SYS-2(2,5): $\Psi \in \sim(2, 5)$ | Large |

**(Q3): TL behavior over underlying physics.** We test both in-domain and out-of-domain physics effects by constructing downstream datasets that systematically deviate from the pre-training dataset. For SYS-1, we sample different ranges for $e$ with varying overlap with the pre-training dataset. Similarly, for SYS-2, we use different ranges of advection-to-diffusion ratio $\bar{\Psi}$ showing different overlap. We highlight these systems (downstream and pre-training) in Tab. 1 for the two PDE systems.

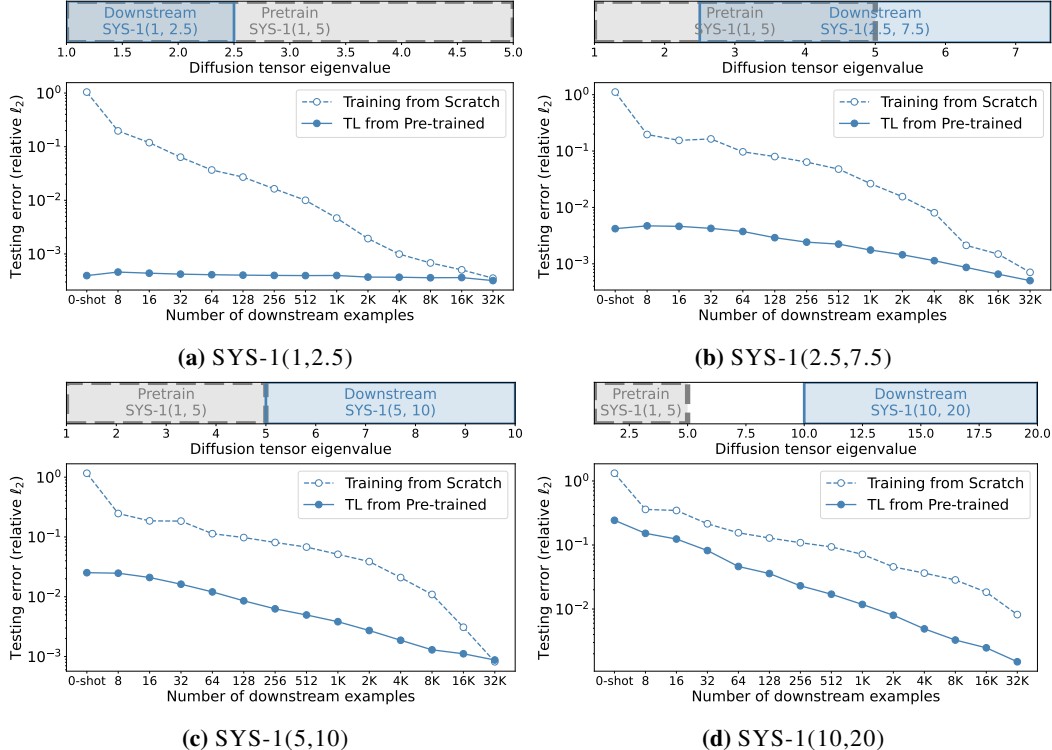

**Figure 4:** *Addressing (Q3). Testing error as a function of downstream examples for different downstream tasks used in* SYS-1. *We show the extent of overlap (signifying distributional shifts) between the pre-trained and downstream dataset at the top using the range of sampled diffusion tensor eigenvalue. For datasets within distribution, zero-shot TL is optimal. As the downstream dataset shifts moderately OOD, the zero-shot learning suffers gradually and is recovered through fine-tuning. This recovery is slower as the distributional shifts increase.*

We repeat our downstream dataset scaling experiments on the different downstream tasks and show the trends for SYS-1 in Fig. 4. In particular, in Fig. 4a, we consider the downstream dataset within distribution of the pre-training dataset (as visualized by the $e$ distribution at the top). We observe excellent zero-shot performance that is unaffected by further fine-tuning. In Fig. 4b, the downstream dataset is shifted mildly OOD. Although the zero-shot performance drops, it still shows low errors, significantly smaller than training from scratch. Further, the performance is improved with few-shot TL up to the point of diminishing returns with large numbers of downstream data examples. With further distributional shift (no overlap) in Fig. 4c, the zero-shot performance suffers more, but with a larger amount of fine-tuning recovers good performance. Finally, for large distributional shifts in Fig. 4d, the zero-shot and few-shot performance is poor, with TL showing relatively high errors, but even here TL improves over training from scratch. In this case, due to larger anisotropy, the system is also harder to emulate and might require more data in general. We repeat this analysis for SYS-2 and observe similar trends across different OOD downstream tasks (see Appendix §B.1).

**(Q4): TL behavior over multiple operators.** We further diversify the pre-training by including examples with different solution operators. We combine the datasets from three PDEs— Poisson's SYS-1(1,5), Advection-Diffusion SYS-2(0.2,1), and Helmholtz SYS-3(1,10) (where the wavenumber $\omega \sim \mathcal{U}(1, 10)$). Here, we have additionally included the Helmholtz PDE, a challenging system due to the highly oscillatory behavior of the solutions (see PDE 2 in Fig. 1 and Fig. A.1, for examples), very sensitive to the range of wavenumbers. When pre-training a single model on this "mixed" dataset, we simply use zero channels for those coefficients that do not exist when using examples from a specific operator. For example, the Helmholtz equation has a diffusion tensor input (identity matrix) with an additional input for the wavenumber but no advection (zero channel), while the Poisson's equation only has a diffusion tensor input and hence we append zero channels to signify no wavenumbers and advection; similarly for Advection-Diffusion. This, effectively, serves

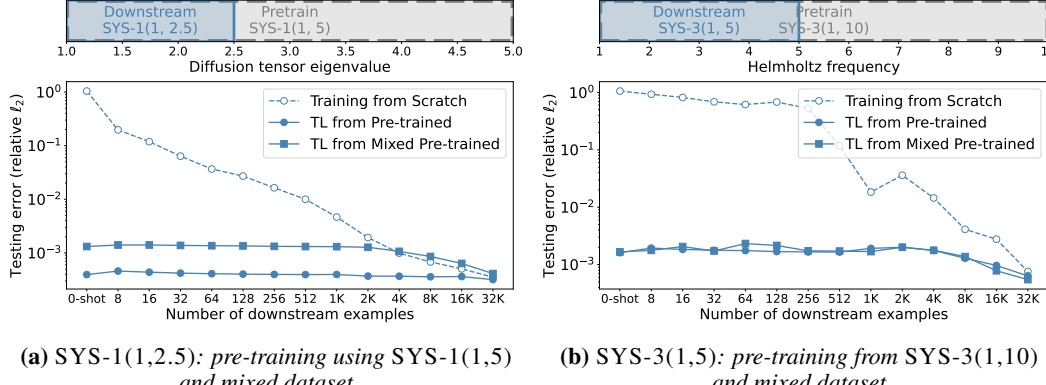

**(a)** $\mathrm{SYS}$-1(1,2.5)*: pre-training using* $\mathrm{SYS}$-1(1,5) *and mixed dataset*

**(b)** $\mathrm{SYS}$-3(1,5)*: pre-training from* $\mathrm{SYS}$-3(1,10) *and mixed dataset*

**Figure 5:** *Addressing (Q4). Testing error as a function of downstream examples for* $\mathrm{SYS}$-1*,* $\mathrm{SYS}$-2*, and* $\mathrm{SYS}$-3 *with fine-tuning from their respective PDE systems and from the mixed dataset (combination of* $\mathrm{SYS}$-1*,* $\mathrm{SYS}$-2*, and* $\mathrm{SYS}$-3*). The model pre-trained on the mixed dataset performs better than training from scratch. More importantly, the same pre-trained model yields low errors on all the downstream PDEs with both zero-shot and task-specific fine-tuning. We show the results for* $\mathrm{SYS}$-2 *and OOD performance in §B.2.*

as selection of the solution operator during the forward pass to predict the solution to the right operator. While more advance techniques such as in-context prompting (from LLMs) exist, here we are interested in understanding if this simple and minimal selection/prompting is sufficient for the model to transfer effectively to downstream tasks. For the downstream tasks, we consider three within-distribution tasks of Poisson's $\mathrm{SYS}$-1(1,2.5), Advection-Diffusion $\mathrm{SYS}$-2(0.2,0.4), and Helmholtz $\mathrm{SYS}$-3(1,5) and show our dataset scaling results in Fig. 5.

The results support our most compelling conclusion: fine-tuning from the mixed dataset retains the substantial performance gains over training from scratch for all downstream tasks. The same model (pre-trained on three different tasks) is useful in all downstream tasks, in both the zero-shot and the fine-tuning settings. This indicates the input coefficient channels are sufficient to prompt the model to predict the correct downstream solution. We show the OOD downstream performance of this model in Appendix §B.2 and observe similar behavior.

## 5 Conclusions

We have provided an extensive analysis of the scaling and transfer behavior of neural operator models on multiple PDE systems. This involved characterizing behavior as a function of model size, downstream dataset size, underlying physics of the downstream tasks in relation to pre-training, the adaptation of these models to multiple downstream PDEs, and how all these behaviors scale with relevant problem and model parameters. Among other things, we have shown that it is possible and beneficial to develop more general SciML models capable of solving multiple tasks with the *same* set of weights, even when downstream tasks involve small-to-moderate distribution shifts relative to the pre-training data. All in all, this demonstrates the potential of the "pre-train and fine-tune" paradigm for SciML problems, paving a path towards building SciML foundation models. Moving forward, many questions remain. These include further exploration of model architectures (balancing expressivity and flexibility), pre-training protocols (including self-supervision components at scale), fine-tuning strategies, and the integration of these within a specific compute (and memory) envelope. There are a number of future directions raised by our work. Our pre-train and fine-tune recipe may need more sophisticated prompting during inference, especially if only the operator form changed between two different PDE systems. Also, we do not look at self-supervision with the PDE loss penalty as a means of large-scale pre-training, and we limit our analysis to 2D spatial systems. Moving to other architectures, larger scales, and more complex PDEs in space-time is a focus of future work.

## Acknowledgements

This work was partially supported by the U.S. Department of Energy, Office of Science, Office of Advanced Scientific Computing Research, Scientific Discovery through Advanced Computing (SciDAC) program, under Contract Number DE-AC02-05CH11231 at Lawrence Berkeley National Laboratory. This research used the Perlmutter supercomputing resources of the National Energy Research Scientific Computing Center (NERSC), a U.S. Department of Energy Office of Science User Facility located at Lawrence Berkeley National Laboratory, operated under Contract No. DE-AC02-05CH11231. We acknowledge gracious support from Google Cloud, Google TRC team, and specifically Jonathan Caton, and Prof. David Patterson. Prof. Keutzer's lab is sponsored by Intel corporation, Intel VLAB team, Intel One-API center of excellence, as well as funding through BDD and BAIR. Amir Gholami acknowledges support from Samsung SAIT and Intel corporations. Our conclusions do not necessarily reflect the position or the policy of our sponsors, and no official endorsement should be inferred.

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

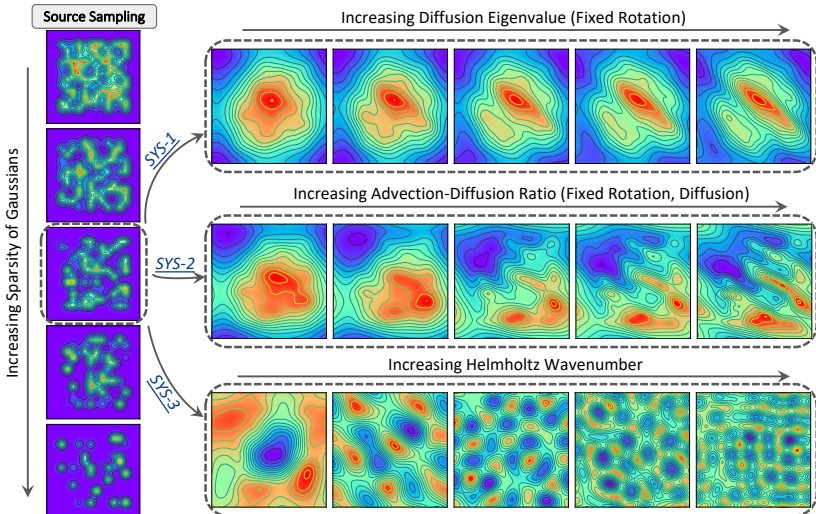

**Figure A.1:** *Visualization of the source function sampling (left) and the effect of certain PDE coefficients (right) on the solutions for the different systems. On the left, as we go down, the sparsity of Gaussians is increasing, leading to more sparse and spread out source functions encouraging heterogeneity in the dataset. For one of these source functions, we apply the different PDE operators with varying ranges of certaino PDE coefficients to illustrate their effect on the solutions. On the top row, for $\mathrm{SYS}$-1 (Poisson's), we show that by increasing the diffusion tensor eigenvalue $e$ (but keeping the direction $\theta$ fixed), we increasing anisotropy and diffusion as we move towards the right. In the middle, we increase the velocity scales for $\mathrm{SYS}$-2 (Advection-Diffusion), but keep the diffusion tensor and velocity direction the same, to demonstrate the increasing competing advection and diffusion processes as we go right. Finally, at the bottom, we show the highly oscillatory behavior in $\mathrm{SYS}$-3 (Helmholtz) as we increase the wavenumber $\omega$. Note the significant differences between the solutions of the different systems.*

## A  Appendix: Additional Details

### A.1  Pre-train and downstream data creation

As described in §3, our sampling strategy involves (i) source functions and (ii) PDE coefficients. We use a numerical discretization of $128 \times 128$ for all our experiments. Some additional details are as follows:

(i) *Source function sampling:* As described in §3, our source functions are a linear combination of $n_g$ radial (Gaussian) basis functions $\{\phi_i(\boldsymbol{x})\}_{i=1}^{n_g}$, where $\phi_i(\boldsymbol{x}) = \phi(\boldsymbol{x} - \boldsymbol{x_i})$ is a Gaussian function centered at grid point $\boldsymbol{x}_i$. Specifically: $f(\boldsymbol{x}) = \sum_{i=1}^{n_g} \phi_i(\boldsymbol{x})p_i$, with $\boldsymbol{p} = \{p_i\}_{i=1}^{n_g}$ as the parameterization vector. The spatial profile controlled by $\sigma$, the standard deviation of the Gaussian function, is preset to a small value $1/32$ (hence 4 times the spatial resolution) to encourage high variability. The spacing between Gaussians is fixed at $2\sigma$. Examples are sampled by uniformly randomly sampling $p_i \sim \mathcal{U}(0, 1)$. We further introduce heterogeneity by controlling the sparsity $s$ of $\boldsymbol{p}$ (we define $s$ as the fraction of number of zero components of $p_i$). Hence, $s = 0.6$, implies only 40% of $\boldsymbol{p}$ is randomly sampled in $\mathcal{U}(0, 1)$ and the rest (60%) are set to zero. Visualizations of this are in Fig. A.1 on the left. As we go down, the sparsity is increased (and hence Gaussians are more sparse and spread apart). In our pre-training and downstream datasets, we sample sparsity levels from 20% to 80% uniformly.

(i) *PDE coefficient sampling:* In $\mathrm{SYS}$-1, we sample diffusion coefficient tensors $\boldsymbol{K} \sim \mathcal{K}(\lambda)$, where $\mathcal{K}$ is a distribution that generates varying scales of anisotropy and spread in the diffusion process: $\boldsymbol{K} = \boldsymbol{R}^{-1}\boldsymbol{D}\boldsymbol{R}$ with $\boldsymbol{D} = \mathrm{diag}(1, e)$ and $\boldsymbol{R} = \mathrm{rot}(\theta)$, where $e$ is an eigenvalue of the tensor that controls the anisotropy and extent of diffusion and $\mathrm{rot}(\theta)$ is a rotation matrix with angle $\theta \sim \mathcal{U}(0, 2\pi)$ that controls the general diffusion direction. We visualize the effect of $e$ in Fig. A.1 (top row). With a fixed $\theta$ (direction of diffusion), we see that with larger $e$, the solution is more anisotropic and diffuse. The sampling of $e$ and $\theta$ in a systematic fashion

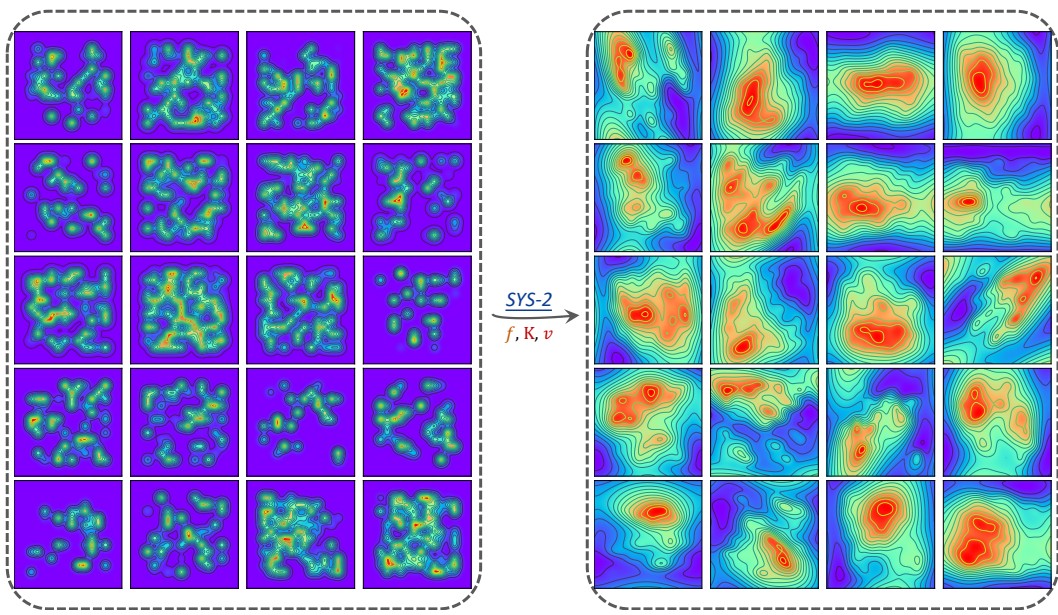

**Figure A.2:** *We illustrate the variability in source function inputs (left) and solution outputs (right) for* SYS-2*, where the velocity direction, scales, and the diffusion tensor direction and anisotropy scales (eigenvalue) are all changed along with the source sampling to produce the input-output pairs for the training dataset.*

quantifies the underlying diffusion physics. In SYS-2, we additionally also sample the velocity vector $\boldsymbol{v}$ direction from $\mathcal{U}(0, 2\pi)$. We define $\Psi = \|\boldsymbol{v} \cdot \nabla u\| / \|\text{div } \boldsymbol{K}\nabla u\|$, the ratio of advection to diffusion to quantify the different processes and $\Psi$ is changed by scaling the velocity. In Fig. A.1 (middle row), we visualize increasing advection (from left to right) when the diffusion tensor is kept the same—observe that the solution changes significantly as $\Psi$ increases and the two processes (advection and diffusion) compete more strongly. In SYS-3, we sample the wavenumber $\omega$ as uniform integers. We visualize the effect of increasing $\omega$ in Fig. A.1 (bottom row) and observe that increasing frequency leads to highly oscillatory behavior. We also underscore the significant differences in the output of the three systems here. For more details, see Appendix §A.1. We visualize the sampling process and resulting solutions with different ranges of underlying physics in Fig. A.1. In Fig. A.2, we show a small sample of input sources and output solutions for SYS-2 to illustrate the variability in our datasets–note here all the variables (advection directions and scales, diffusion directions and scales, source functions) are changed to create the full dataset.

For each PDE system, we use spectral methods to compute the ground truth solution of the PDE. For example, for SYS-1:

$$-\text{div } \boldsymbol{K}\nabla u = f, \tag{4}$$

the solution is:

$$u = \mathcal{F}^{-1}\Big(\frac{-1}{-(k_x^2 k_{11} + k_y^2 k_{22} + 2k_x k_y k_{12})}\mathcal{F}(f(\boldsymbol{x}))\Big), \tag{5}$$

where $\mathcal{F}$ is the Fourier transform, $k_x, k_y$ are the frequencies in the Fourier domain, and $k_{11}, k_{22}, k_{12}$ are the diagonal and off-diagonal coefficients of the diffusion tensor $\boldsymbol{K}$. The solution is unique given zero-mean source functions. Similar solutions can be derived for the other systems.

## A.2 Model architecture

We describe the architecture details of the FNO here. The central building block of FNO learns a kernel integral operator parameterized by a function operating in Fourier space. By composing several of these blocks together in sequence, the FNO can learn to approximate the solution operator for complex PDEs. For Euclidean systems discretized at uniform resolution (e.g., in 2D with inputs and solutions having dimension $\mathbb{R}^{h \times w}$), the implementation can be made efficient by using the

Fast Fourier Transform (FFT), denoted by $\mathcal{F}$ with inverse FFT $\mathcal{F}^{-1}$. We refer the reader to the original FNO paper [30] for details, but briefly summarize the basic mechanism, to highlight how the model complexity depends on key hyperparameters. The inputs to the 2D FNO are spatial functions discretized at this resolution $h \times w$ that represent sources, PDE coefficients, ICs/BCs of the PDE system–each of these is represented as a separate channel, leading to input tensors in $\mathbb{R}^{h \times w \times c}$, with $c$ input channels. As mentioned before, we use the resolution $h \times w = 128 \times 128$. Given this input tensor $A \in \mathbb{R}^{h \times w \times c}$ ($c$ input channels representing the PDE inputs), the FNO first projects $A$ into a tensor $X \in \mathbb{R}^{h \times w \times d}$ with embedding dimension $d$, which is passed through a series of FNO blocks. For a given block $l \in \{1, .., L\}$ with input $X^l \in \mathbb{R}^{h \times w \times d}$ the output at spatial index $(i, j)$ is computed as

$$X^{l+1}_{(i,j)} = \sigma(W_l X^l_{(i,j)} + \mathcal{F}^{-1}[\mathcal{K}_l(\hat{X}^l)]_{(i,j)}), \tag{6}$$

where $\sigma$ is a pointwise nonlinear activation function, $W_l \in \mathbb{R}^{d \times d}$ is a learnable weight matrix, which performs pointwise linear transformation, and $\hat{X}^l = \mathcal{F}(X^l) \in \mathbb{C}^{h \times w \times d}$ are complex-valued Fourier coefficients output by the FFT. The transformation $\mathcal{K}_l$ in Fourier space is parameterized by complex weights $\Phi^l \in \mathbb{C}^{d \times d \times m_1 \times m_2}$ according to

$$\mathcal{K}_l(X^l)_{(k_1,k_2)} = \Phi^l_{(k_1,k_2)} X^l_{(k_1,k_2)}, \tag{7}$$

for pairs of Fourier frequencies/wavenumbers $k_1 \in \{1, ..., m_h\}$ and $k_2 \in \{1, ..., m_w\}$. The hyperparameters $m_h$ and $m_w$ control the "mode cutoff", beyond which Fourier modes are ignored by $\mathcal{K}$, and they have a theoretical maximum of $m_h = h/2$ and $m_w = w/2$, i.e., the Nyquist limit along each spatial dimension. In practice, the mode cutoffs are a key hyperparameter controlling model complexity along with the embedding dimension $d$, and they are often tuned to be less than the Nyquist limit to prevent overfitting and accelerate training. For square problems ($h = w$), a symmetric truncation is adopted such that $m = m_h = m_w$. Thus, the per-layer parameter count is quadratic in both $m$ and $d$, dominated by the complex weights $\Phi$. The above ($d$ and $m$) hyperparamters are the focus of our exploration in the model scaling experiments.

### A.3 Training details and code open-source

For training, we use the Adam optimizer with a cosine learning rate decay schedule. All models are trained for 500 epochs with the best model saved at lowest validation loss. We tune batch size and initial learning rates using a grid hyperparameter search, and train every model using 4 NVIDIA A100 GPUs (using standard data-parallelism) on the Perlmutter supercomputer. A full training run (on all $32K$ examples used for pre-training) completes in around 2.5 hours. To evaluate model performance, we use the mean relative error between predicted and target solution, defined as: $\mu_{\ell_2} := 1/N \|u - u_0\|_2 / \|u_0\|_2$, where $u$ is the predicted solution, $u_0$ is the target solution, and $N$ is total (fixed) number of testing (around $4K$) examples. We open source our model training and data generation code at [1].

### A.4 Hyperparameter tuning

For studying the relationship between model and dataset size, we perform a simple grid-search over 5 different learning rates for each combination of model and dataset size. This is important because different dataset sizes may demand different hyperparameter values—for instance, we observe that for very small dataset sizes (small number of downstream examples), the tuned learning rates are significantly smaller (especially for zero- and few-shot transfer learning) to help mitigate over-fitting and optimization difficulties at that scale, whereas larger learning rates perform better as model and dataset sizes increase. Hence, for any result (including the model scaling), these learning rates are tuned with the best values picked from the validation set metrics.

### A.5 Input normalization

As mentioned in §3, a key aspect of our training is the input normalization (for any PDE system). We first normalize every training example source and coefficient inputs with respect to a reference source value defined as the median source norm over the training dataset, i.e., $f_{\text{ref}} := \text{median}_i(\|f_i\|_2)$ where $i$ are the training examples. Hence for any example $f_i$, we first compute $\|f_i\|_2$ and normalize $f_i$ and all coefficients with the relative norm $\|f_i\|_2 / f_{\text{ref}}$. First, this ensures the source norms are within a reasonable scale range. Additionally, it implies that scaling both coefficients as well as source

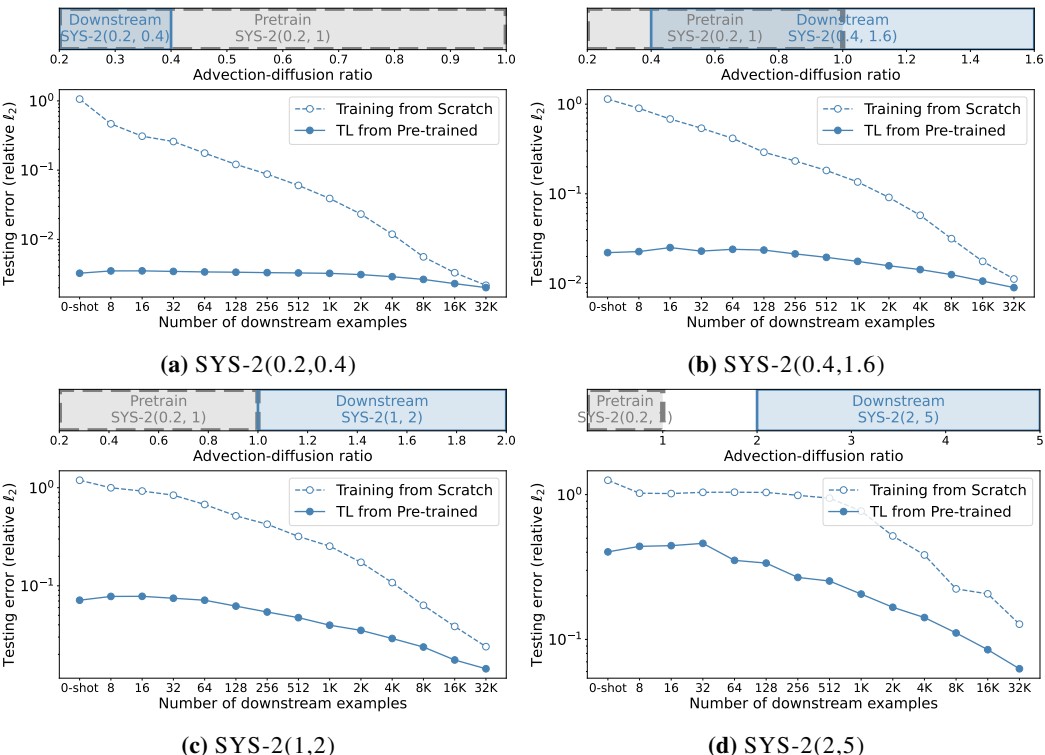

**Figure A.3:** *Addressing (Q3). Testing error as a function of number of downstream dataset examples for different downstream tasks used in* SYS-2. *We show the extent of overlap between the pre-trained and downstream dataset at the top using the range of advection-diffusion ratios of the two datasets. Similar to* SYS-1*, we observe good zero-shot performance that gradually decreases with distributional shifts but can be recovered through few-shot learning. With large shifts, the number of examples needed to reach desired error levels also increases.*

functions of the inputs by a constant yields the same inputs to the neural operator. For example: the inputs $f$ and $\boldsymbol{K}$ to the Poisson's equation (SYS-1) are equivalent to the inputs $10f$ and $10\boldsymbol{K}$—both have the same solution function. The above normalization makes sure that these two input pairs are equivalent, since $10f$ and $10\boldsymbol{K}$ simply get normalized by 10 before passing to the network. Then, to ensure the coefficient inputs are comparable scales and within a reasonable range, the coefficient values (that typically have very different scales depending on the physical process, up to $100\times$ differences) are normalized by constant reference values pre-computed across the training dataset as the median values of the coefficients. We observe that without normalization the performance of FNO can be quite poor, especially in cases with multiple channels representing different physical scales (such as in advection-diffusion SYS-2).

## B  Additional Results

### B.1  TL behavior over underlying physics and model scaling

We show the TL performance under physics shift (**Q3**) for SYS-2 in Fig. A.3. Similar to SYS-1, we observe excellent TL performance for in-distribution tasks (see Fig. A.3a) that is independent of the downstream dataset size. As we systematically go OOD, we continue to observe good performance gains with TL (both zero-shot and few-shot) until we are significantly OOD as in Fig. A.3d.

In both SYS-1 and SYS-2, we find that TL from pre-trained models outperforms training from scratch with the number of downstream examples required to reach a given accuracy being orders of magnitude smaller with TL. To quantify this in detail, we estimate the amount of "data savings" for TL by interpolating the equivalent number of examples needed to reach a given model error when training from scratch. We plot these "from-scratch" data requirements as a function of number of

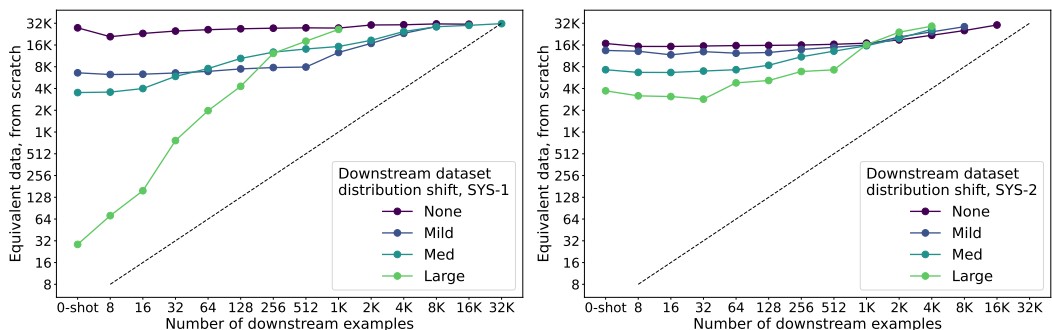

**Figure B.1:** *Addressing (Q3). Equivalent data needed to reach the accuracy of TL when training from scratch, for* SYS-1 *and* SYS-2 *distribution shifts defined in Table 1. Data from experiments where TL outperforms even the best (i.e., largest dataset) from-scratch models is not plotted.*

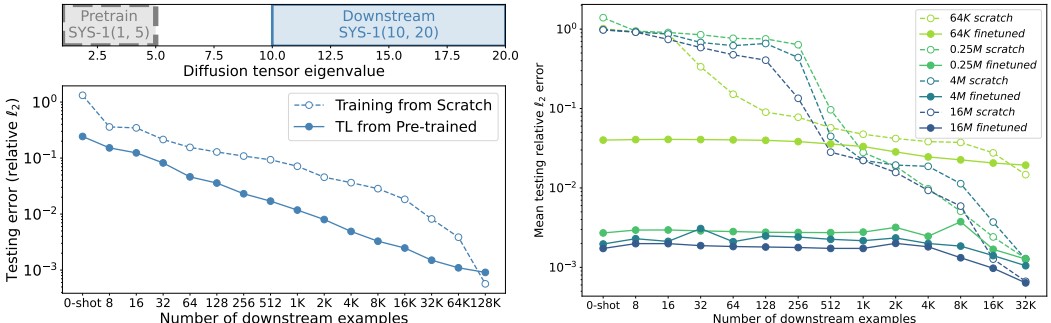

**Figure B.2:** *Some additional results: (left) for extreme shifts and challenging systems, the number of examples to reach a desired accuracy increases. For example, for TL to* SYS-1(10,20)*, the training from scratch and TL curves intersect at 128K examples. (right) model parameter scaling for TL to* SYS-3(1,5) *showing similar trends as other systems.*

TL fine-tuning examples in Fig. B.1. In these plots, any points above the dashed black line indicate an improvement over training from scratch; points for TL experiments which outperform even the best from-scratch models (which use a maximum of $32K$ training examples) are not plotted. We observe that even as the downstream dataset distribution shift moves from "None" to "Large" (as defined in Table 1), the TL models consistently reach the performance of from-scratch models which would otherwise require thousands to tens of thousands of training examples. One exception is the largest OOD shift for SYS-1, where we observe smaller (but still substantial) advantages for the TL approach in zero- and few-shot learning. We note that characterizing distribution shifts consistently for different systems is challenging; this result suggests perhaps the "Large" distribution shift for SYS-1 is a more significant shift than that of SYS-2.

Finally, we include two additional results: For extreme shifts such as SYS-1(10,20), we note that the errors are higher as its is a more challenging system (high anisotropy in diffusion, in this case) and hence needs more data to achieve lower errors. In Fig. B.2 (left) we show that the training from scratch and TL from pre-trained model curves converge at a much larger dataset of 128K examples. We also include the model scaling experiments for a Helmholtz system SYS-3(1,5) in Fig. B.2 (right).

## B.2 TL behavior underlying multiple operators

We show additional results for TL behavior over multiple operators in Fig. B.3. Each row represents a PDE system with in-distribution (left) and OOD (right) downstream tasks. We observe that the mixed pre-trained model is able to retain TL gains as the system-specific pre-trained model over all the systems. Hence, a single model is now adapted to three different PDE systems in downstream fine-tuning. We also note SYS-3 OOD results—we observe that for the Helmholtz, going even moderately OOD, affects both TL and training from scratch performance. This PDE system is

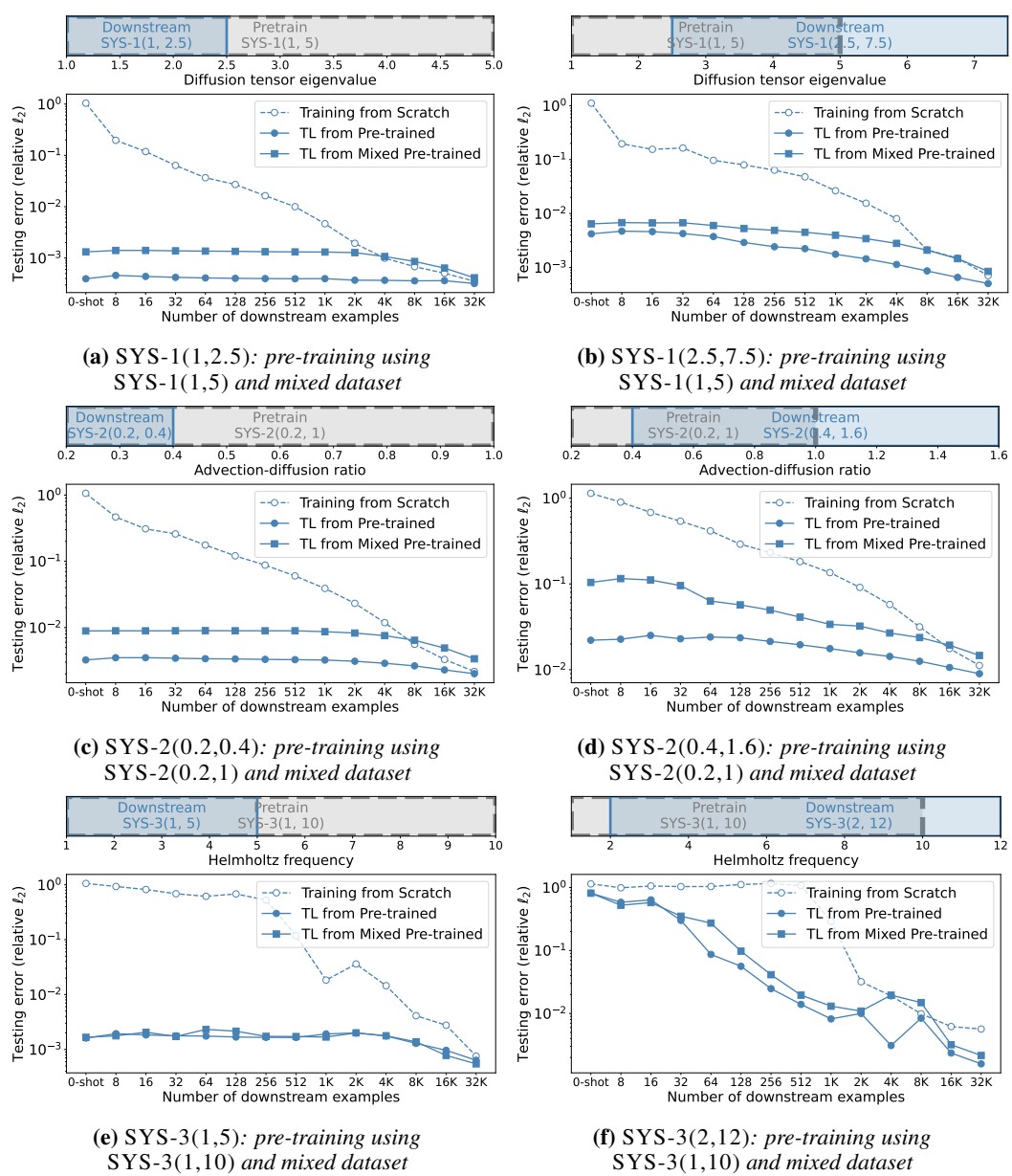

**(a)** SYS-1(1,2.5)*: pre-training using* SYS-1(1,5) *and mixed dataset*

**(b)** SYS-1(2.5,7.5)*: pre-training using* SYS-1(1,5) *and mixed dataset*

**(c)** SYS-2(0.2,0.4)*: pre-training using* SYS-2(0.2,1) *and mixed dataset*

**(d)** SYS-2(0.4,1.6)*: pre-training using* SYS-2(0.2,1) *and mixed dataset*

**(e)** SYS-3(1,5)*: pre-training using* SYS-3(1,10) *and mixed dataset*

**(f)** SYS-3(2,12)*: pre-training using* SYS-3(1,10) *and mixed dataset*

**Figure B.3:** ***Addressing (Q4).*** *Testing error as a function of downstream examples for* SYS-1 *(top),* SYS-2 *(middle), and* SYS-3 *(bottom) with fine-tuning from their respective PDE systems and from the mixed dataset (combination of* SYS-1*,* SYS-2*, and* SYS-3*). In each row: on the left, the downstream task is in-distribution and, on the right, it is OOD. The model pre-trained on the mixed dataset performs better than training from scratch. More importantly, the same pre-trained model yields low errors on all the downstream PDEs with both zero-shot and task-specific fine-tuning. We also note that* SYS-3 *is a particularly challenging system that shows larger performance drops as we go OOD—the mixed pre-training still retains the same gains as the task-specific pre-trained model.*

particularly challenging and is very sensitive to the input wavenumber. Slight changes introduce more oscillatory features in the solution, pushing the downstream task OOD easily. However, we note that both the mixed pre-training and pre-training from a Helmholtz dataset show similar performance even on the OOD task.

## B.3 Sensitivity to random seeds

To quantify the variability of both training from scratch and performing TL from the pre-trained model, we repeat the "Med" OOD shift experiment for SYS-1 and SYS-2 (see Table 1) 5 times with different random seeds and record the testing error for each trial. The resulting distributions indicate how sensitive each approach is to the random data shuffle used when training on the downstream dataset. We plot the mean testing error along with $1^{st}$ and $3^{rd}$ quartiles for each downstream dataset size in Fig. B.4. We observe small sensitivity for both TL and training from scratch. Further, not surprisingly, we observe that the variability across random seeds is generally larger when training from scratch. We note that the small variability in TL is difficult to see due to the log-spaced y-axis.

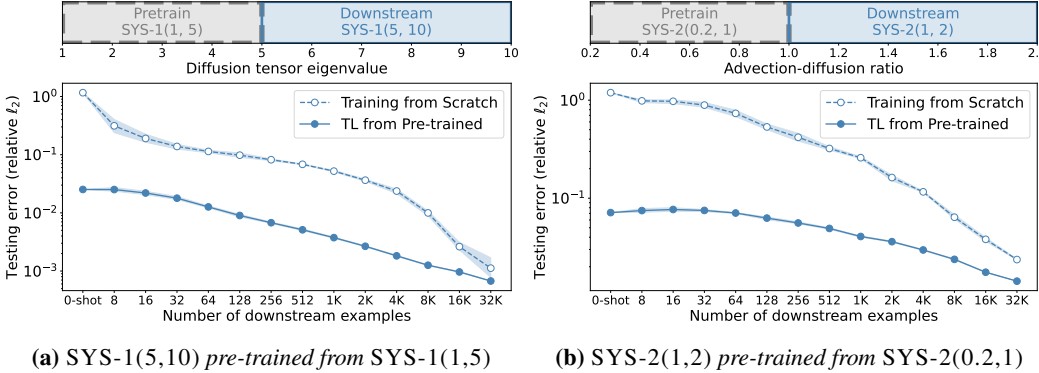

**(a)** SYS-1(5,10) *pre-trained from* SYS-1(1,5)  **(b)** SYS-2(1,2) *pre-trained from* SYS-2(0.2,1)

**Figure B.4:** *Testing error as a function of downstream examples for* SYS-1 *and* SYS-2*, aggregated over 5 trials with different random seeds for each experiment. The dots indicate the mean testing error at each downstream dataset size, and the shaded region represents the spread between the $1^{st}$ and $3^{rd}$ quartiles. We observe small variance that is slightly larger when training from scratch compared to TL from the pre-trained model.*

