# OpenReview forum: "Towards Foundation Models for Scientific Machine Learning: Characterizing Scaling and Transfer Behavior"
_NeurIPS.cc/2023/Conference — NeurIPS 2023 poster_

### Official Review · Reviewer_QnfD · 2023-07-05

**Soundness:** 3 good
**Presentation:** 4 excellent
**Contribution:** 3 good
**Rating:** 7
**Confidence:** 3

**Summary:**

This work explores and demonstrates how a "pretraining+finetuning" paradigm may be leveraged for neural networks applied to PDEs. Specifically, for three different systems governed by distinct PDEs, the authors study how different factors influence performance: (1) downstream dataset scale; (2) model scale; (3) OOD-ness of the downstream data; (4) pretraining on systems governed by different physics. The authors find that pretraining systematically improves performance, delivering significant performance gains in limited downstream data settings -- which are particularly relevant to SciML, which is often more data-constrained than NLP.

**Strengths:**

* **S1.** This work could be very significant for the community: it explores important topics for SciML, presents extensive strong results, and is easy-to-follow, well-structured, and pleasant to read. The paper is also extensively detailed (regarding data setup, architectures, etc.).

* **S2.** The authors study a variety of scenarios (OOD-ness, downstream dataset size, etc.) and perform appropriate comparisons with a "from scratch" baseline. The identified strength of the approach (in the low downstream data regime) answer particularly well challenges in the field.

* **S3.** The authors open-source their code, which will help with propagating their work, increasing its impact, and enabling better reproducibility.

**Weaknesses:**

* **W1. The experiments could be more systematic across the three systems proposed.**
    * **W1.1.** The SYS-3 setup is not featured for downstream dataset scaling & model scaling; it would be valuable to see it in these scenarios to further validate the method, especially since it is described as the most challenging setting.
    * **W1.2.** The behaviour over multiple operator could be better explored: it is lacking a comparison with SYS-2, and is done only on a very "in-domain" case (while Figure 1 for instance is done with disjoints domains) thus causing the n-shot behaviour to be flat.
   * **W1.3.** (minor) In Figure 4d) (with extreme OOD on SYS-1), the from scratch behaviour is different, never catching-up with the TL case. This diverges from the trend and warrants some explanations.

* **W2. Some of the framing around large language models is inaccurate.**
   * **W2.1.** l25 "An important aspect of the trained foundation model is the notion of emergence—the model is able to perform tasks seemingly different than those for which it was trained by leveraging shared features across the training tasks." It is misleading to characterise emergence with this sentence. As discussed in Wei et al., 2022 emergence is more accurately described by "abilities that are not present in smaller-scale models but are present in large-scale models".
   * **W2.2.** The few-shot terminology can be confusing in some cases. For large language models, zero/few-shot is synonymous with in-context learning. Mentioning few-shot alone simply means prompting the model. Here, in this work, it systematically means finetuning the model on these few shots.
   * **W2.3.** l227 "This is motivated by observations in NLP that after a critical model size, we can expect significantly better adaptation performance" it's unclear what the authors refer to here, as finetuning efficiency in NLP is closely related to sample efficiency and is known to increase smoothly with model size (Kaplan et al., 2020).
   * **W2.4.** (minor) l145 the focus on GPT models is a bit odd: finetuning applies to all classes of modern language model, and in fact the GPT (i.e., causal decoder-only) models are somewhat unique in their ability to not require finetuning. Conversely, they do not perform as well for fine-tuned tasks as other models: see What Language Model Architecture and Pretraining Objective Work Best for Zero-Shot Generalization? Wang et al., 2022.
   * **W2.5.** (minor) l25 the references to scaling laws are missing a citation for Hoffmann et al., 2022, which is the reference work at this time and which highlighted the need for joint scaling of data and model size instead of model size alone predominantly.

* **W3.** Smaller nits: l44 "TL" to refer to transfer learning as not been introduced as an acronym.


**Questions:**

Overall, this is an interesting paper addressing important topics for the community, with the potential to be highly valuable. It is however slightly held back by some concerns around the showcased experiments. Addressing these would further validate the wide applicability of the approach, confirming its significance and increasing the quality of the work. Accordingly, I am rating this work as a **Weak Accept (6)** but would be willing to increase my rating should my concerns/questions be addressed.

**EDIT: following rebuttal, I have updated my score to an Accept (7)**.

* **Q1.** (W1.1.) Could the authors provide results on SYS-3 for downstream dataset scaling and model scaling?

* **Q2.** (W1.2.) Could the authors provide additional results for Figure 5./Q4, demonstrating the approach on SYS-2, and across all systems but OOD? (like in Figure 2).

* **Q3.** (W1.3.) Could the authors comment on why the extreme OOD in Figure 4d) behaves differently?

* **Q4.** (W2.3.) Could the authors clarify their statement on l227/provide a reference?

* (W2.2.) As a suggestion, to avoid confusion for readers, I would recommend systematically saying "few-shot finetuning" and not using "few-shot learning": l73 "few-shot TL" could be unclear, l132 "few-shot learning" --> "few-shot finetuning", l221 "few-shot learning" --> "few-shot finetuning"

**Limitations:**

The authors discuss some of the limitations of their work in the conclusion, but I would encourage a more critical look regarding when the proposed method could fail.

---

> ### Author Rebuttal · Authors · 2023-08-09
>
> We thank the reviewer for their detailed feedback, thorough assessment and valuable comments/suggestions regarding our paper.
>
> ### R4-1: W1: Additional results for SYS-3/SYS-2 to be more systematic
> That is a good point. We should mention that SYS-3 (helmholtz) was chosen specifically as a system to stress-test our findings rather than just a third PDE system to boost our results. It is a well-known system in scientific computing that demonstrates very challenging numerical issues while solving. This system shows dramatic frequency spectrum changes as coefficients change and can also exhibit ill-conditioned behavior where the solution norm increases by a large amount when the coefficient changes a little, making it particularly hard for a neural network model to continuously model (some examples in Appendix Fig A.1), and even harder to generalize. Our goal here is to investigate if such difficulties preclude us from building a model trained on multiple PDEs. We agree that this motivation can be made more explicit and expanded with more detail in l166 as well as in the results. Specifically, we address the weaknesses mentioned in the review below:
>
> **W1.1/Q1**: The downstream dataset scaling can be seen for SYS-3 in Fig. 5.b for in-distribution and Appendix Fig. B.3.f for OOD (the 'circle' labels are the downstream dataset scaling). As we can see, the in-distribution SYS-3 shows similar properties, whereas the OOD performance is poorer than the other systems (the model shows high errors for coefficients that are OOD, reinforcing the challenging nature of this system). For model scaling, we agree with the reviewer to include the SYS-3 scaling result as well. We repeat our experiments for SYS-3 in-distribution and will include the result in Fig 1 of the attached PDF in the global response in the paper.
>
> **W1.2/Q2**: The SYS-2 results as well as OOD results are in Appendix Fig. B.3. Due to space constraints, we were unable to place this figure in the main text. We observe that mixing datasets (multiple operators) and fine-tuning the pre-trained model still demonstrates gains across all the systems: see Appendix Fig. B.3 for all three systems, evaluated both in- and out-of-distribution. Even for the OOD SYS-3 which shows poor performance, there is no additional degradation from using a pre-trained model trained on the mixed (multiple operators) datasets.
>
> **W1.3/Q3**: The extreme OOD case is a more challenging system with large anisotropy in diffusion, so we expect the network to have more difficulty modeling this case. We note that the errors on the y-axis are increasing (for the same number of downstream examples) as we move more OOD and to more challenging systems. In other words, larger amounts of data might be needed to obtain lower errors. We add two more data points with 64K and 128K dataset sizes for the extreme OOD case and observe that at 128K, the curves from training from scratch and fine-tuning converge as shown in Fig 2 of the attached PDF in the global response.
>
> Our interpretation of this result is that for more extreme datasets as this, more samples are required to reach a given accuracy; we also note that our method of generating OOD data and measuring the extent of OOD the samples are, is based on the diffusion tensor eigenvalues (and other coefficients for the other two systems), and the categories of “Mild/Medium/Large” distribution shifts in Table 1 are qualitative categories we use to help the reader understand each dataset easier; the exact point at which from-scratch training intersects the error of a pre-trained model will not necessarily always occur in the same place for all systems and distribution shifts.
>
> ### R4-2: W2: Framing of language around NLP models
> We mainly include examples of LLMs as a regime of DL where foundation models have been shown to be successful and use them to frame our high-level discussion points. However, we agree the language used could be improved and clarified and deeply appreciate the reviewer’s suggestions and feedback here.
>
> **W2.1**: That is right, we will fix this in the revised version of the paper and clarify based on the reviewer's suggestion.
>
> **W2.2**: We will change the terminology everywhere to “few-shot finetuning” as-per the reviewer’s suggest to remove this confusion. We will also clarify that this is what we mean by “few-shot TL”.
>
> **W2.3/Q4**: With this sentence, we simply meant that larger models show better testing performance given a dataset size and refer to Fig 2 of Kaplan el al. (2020). We agree that the wording is quite confusing and will clarify to the above sentence.
>
> **W2.4**: We will clarify this in the text. We mostly mentioned GPT as a popular example of a foundation model but we see how this can cause confusion.
>
> **W2.5**: We will add this reference to l25.
>
> **W3**: We will fix this reference.

---

> > ### Comment · Reviewer_QnfD · 2023-08-16
> >
> > I would like to first thanks the authors for providing a rebuttal to each review and for providing additional results.
> >
> > The authors have addressed my concerns appropriately, and I do not see significant issues sticking out from other reviews. Accordingly, I have updated my score to an **Accept (7)**.

---

### Official Review · Reviewer_o2Fy · 2023-07-05

**Soundness:** 3 good
**Presentation:** 4 excellent
**Contribution:** 3 good
**Rating:** 6
**Confidence:** 3

**Summary:**

This paper investigates the finetunability, transfer, and generalization properties of a foundational model applied to the field of scientific machine learning, as functions of finetuning dataset size, pretrained model size, and value of physics parameters governing the equations of the system.

**Strengths:**

The paper is well-written and well-organized, easy to follow and understand. The authors present their method clearly, and they provide a detailed evaluation of their method, with findings clearly highlighted and supported by strong empirical evidence. The experimental hypothesis are clearly stated up front and verified throughout the paper, with sound methodology and experiment design.

**Weaknesses:**

A weakness, also acknowledged by the authors at the end of Section 3 -- but without much contextualization --, is that they only explore the questions raised in the paper in the Fourier Neural Operator (FNO) architecture, and present it as a prototypical SciML architecture. Section 3 explains that this is a good choice for problems with periodic boundary conditions like the ones explored in this paper. However, it's unclear whether any of these results would hold for other types of OOD problems or other types of architectures. Therefore, without more contextualization, it's unclear whether these findings are generalizable and would allow the community to build on top of these findings, or whether they are anecdotal, and would only hold for the architecture and type of problems explored in this work.

Another weakness is that some of the findings, as currently present, merely seem to provide additional experimental evidence for notions that are already widely accepted in the community, such as the fact that, in the low downstream data regime, pretraining a model and then finetuning it on this data is preferable (and yields higher performance) than training a model on this data from scratch, or that finetuning small capacity models plateaus to a certain irreducible error even if data continues to increase.

Finally, while the authors acknowledge that a concurrent paper covers the alternative approach of in-context learning (as opposed to finetuning), I would have expected the authors of this paper to compare against that alternative, as opposed to simply benchmarking against training from scratch, which is a much less tight comparison. I think this paper would benefit from the inclusion of that baseline.

**Questions:**

Small suggestion: please provide common examples of physical systems/phenomena described by the PDEs explored in this work.

**Limitations:**

While disclosing the limitation of this analysis being limited to FNO, the author could be more forthcoming about the consequences of this limitations and the applicability, at a broader level, of these findings. Overall, however, the authors do an excellent job at not overstating their claims and clearly explaining the experimental setup and application domain this paper applies to.

---

> ### Author Rebuttal · Authors · 2023-08-09
>
> We thank the reviewer for their positive feedback and constructive criticisms and address specific questions and suggestions below:
>
> ### R3-1: Exploring only one architecture:
> We agree with this limitation of using just the FNO and make this point in the paper. We are happy to expand on this more. This concern was also raised in R2-1 and we copy our responses below here:
> We call out this limitation in Sec. 1, 2 and 5. We choose the FNO partly because we consider PDE systems with periodic boundary conditions in this work, which FNO is well adapted for, and primarily due to its status as one of the more widely-used architectures in SciML. Since its introduction (in which all experiments focused on PDE systems), FNO and its variants have been applied to a variety of challenging SciML applications including large physical systems like global weather forecasting (https://arxiv.org/abs/2202.11214), multiphase flow (https://arxiv.org/abs/2109.03697), and seismology (https://arxiv.org/abs/2108.05421) to name a few. However, we would like to note our intentional choice to (1) only evaluate one architecture and (2) use the FNO, in particular. The primary reason for experimenting with a single architecture was to make the scope of the study more feasible from a compute and analysis perspective – as mentioned in Sec. 1 l45, the reality is that the domain of SciML for PDEs is far from converged on a single dominant architecture and the different architectures in circulation can have very different properties. This is partly due to the great diversity of physical scales, behavior, and geometries found in SciML applications; that is why we wanted to focus on some critical open questions (dataset diversity/scale, model scaling, and generalization across physics) and fix some of the other variables (model architecture, pretrain/fine-tune strategy) in order to enable a systematic study within a realistic compute budget. However, we do agree with the reviewer that our study should be repeated with different architectures, and comparative analysis between both existing popular SciML architectures (FNO, DeepONet, etc) as well as more conventional ones successful in general ML applications (e.g. ViTs and related variants) would be useful for the community.
>
> ### R2-2: Results being more or less consistent with findings in CV/NLP domains:
> We agree the differences in the PDE domain compared to CV/NLP should be called out more clearly in the paper, both in the introduction and in the results. A similar concern was expressed in R1-2 and we copy our response here:
> A major difference in the PDE domain is the diversity of scales, features, and solution behaviors across PDEs. For example, while OOD shifts in CV/NLP typically involve things like different color distributions or semantic contexts, in a PDE system, it is possible to get significantly different behavior of the PDE solution as physics parameters (and/or initial and boundary conditions) change. These can manifest as new features (for example, increased emphasis on very different frequency scales of the solution causing significant shifts in detail, sharpness, and other image attributes to be visible in the PDE solution) and depend on the physics coefficients in a highly non-linear fashion. We visualize this diversity and sensitivity for our PDE systems in Appendix Fig A.1 to highlight the above point. In general, this diversity can be far greater in larger scientific systems (such as bifurcation in fluid flows where the dynamics can change dramatically with relatively small changes to the PDE inputs) and further amplified by ill-conditioned or numerically stiff systems. From a deep learning perspective, the dense prediction (and TL) of solutions exhibiting these characteristics with input numerical scales that can span orders of magnitude becomes a challenging task. One specific example of this that we mention in section 3 and appendix A.5 is our normalization strategy, which our results are quite sensitive to.
>
> ### R3-3: Examples of scientific systems:
> This is a very good point and we will add examples for the three systems we considered that are fundamental in many science applications. The Poisson’s equation and its variants are used in electrostatics (potential fields caused by an electric charge distributions), solid mechanics (deformation fields through external forces), fluid mechanics (pressure distributions in incompressible flows), amongst other applications in astrophysics, chemistry, mechanics, image processing. The advection-diffusion and its variants are used in fluid mechanics (transport of particles, energy, and other physical quantities), atmospheric physics (wind forecasting), semiconductor physics, solid mechanics (for example, eulerian formulations of cancer growth models). The Helmholtz system, is a particularly challenging system, and has applications in electromagnetics and acoustics (propagation of waves) and seismology (earthquake simulations).
>
> ### R3-4: Comparisons to in-context learning:
> We agree with the reviewer that the concurrent work that appeared with our paper in introducing in-context learning is an interesting methodology to compare against and repeat our systematic analysis. However, that work (Yang et al. 2023, https://arxiv.org/pdf/2304.07993.pdf) focuses on much simpler systems (ODEs and 1D) and at much smaller scales and is also more suitable for cases where only limited amount of data is available for a target problem (3-5 in the case of that paper). Generally, fine-tuning approaches should perform better when more data is available for a target domain. As for the small data regime, however, we agree that in-context learning could be a very suitable method to apply foundation models. This is a direction that we are actively pursuing and we will clarify this point in the revised version of the paper.

---

### Official Review · Reviewer_9JLh · 2023-07-07

**Soundness:** 4 excellent
**Presentation:** 4 excellent
**Contribution:** 3 good
**Rating:** 6
**Confidence:** 2

**Summary:**

This paper analyzes the transfer performance of the pre-trained foundation model for scientific machine learning applications described by PDEs. A wide range of settings is considered: various sizes of downstream datasets, pretrained model, out-of-distribution data, as well as multi-tasks and applications. Conclusions, though not necessarily surprising, are drawn under these settings.

**Strengths:**

- To the best of the reviewer's knowledge, this is a relatively comprehensive analysis of pre-training and transfer learning regimes over a diverse set of physical systems.

- The presentation is clear and easy to follow, and the experiments are comprehensive and convincing.

**Weaknesses:**

- For the experiments, more network architectures can be considered.

- The findings provided by the paper are more or less consistent with NLP and cv tasks. This is not necessarily a drawback, but the paper would be stronger if it provides some findings that make Scientific Machine Learning different from other well-explored AI tasks.

**Questions:**

We can observe from the figures that the performance gap is reduced when more fine-tuning examples are observed. This is not surprising, and the advantage of the pre-trained foundation model is indeed when fine-tuning examples is less. In this case, the performance is worse. I am wondering how to interpret the error, and at what magnitude, the error becomes acceptable for these problems.

**Limitations:**

The author adequately addressed the limitations.

---

> ### Author Rebuttal · Authors · 2023-08-09
>
> We appreciate the positive feedback and constructive criticisms from the reviewer. While we are happy there are no major concerns, we address some of the specific suggestions below.
>
> ### R2-1: Exploring only one architecture:
> We agree this is a major limitation of our work, and we call this out in Sec. 1, 2 and 5. We choose the FNO partly because we consider PDE systems with periodic boundary conditions in this work, which FNO is well adapted for, and primarily due to its status as one of the more widely-used architectures in SciML. Since its introduction (in which all experiments focused on PDE systems), FNO and its variants have been applied to a variety of challenging SciML applications including large physical systems like global weather forecasting (https://arxiv.org/abs/2202.11214), multiphase flow (https://arxiv.org/abs/2109.03697), and seismology (https://arxiv.org/abs/2108.05421) to name a few. However, we would like to note our intentional choice to (1) only evaluate one architecture and (2) use the FNO, in particular. The primary reason for experimenting with a single architecture was to make the scope of the study more feasible from a compute and analysis perspective – as mentioned in Sec. 1 l45, the reality is that the domain of SciML for PDEs is far from converged on a single dominant architecture and the different architectures in circulation can have very different properties. This is partly due to the great diversity of physical scales, behavior, and geometries found in SciML applications; that is why we wanted to focus on some critical open questions (dataset diversity/scale, model scaling, and generalization across physics) and fix some of the other variables (model architecture, pretrain/fine-tune strategy) in order to enable a systematic study within a realistic compute budget. However, we do agree with the reviewer that our study should be repeated with different architectures, and comparative analysis between both existing popular SciML architectures (FNO, DeepONet, etc) as well as more conventional ones successful in general ML applications (e.g. ViTs and related variants) would be useful for the community.
>
> ### R2-2: Results being more or less consistent with findings in CV/NLP domains:
> We agree the differences in the PDE domain compared to CV/NLP should be called out more clearly in the paper, both in the introduction and in the results. A similar concern was expressed in R1-2 and we copy our response here: A major difference in the PDE domain is the diversity of scales, features, and solution behaviors across PDE coefficients as well as systems of interest in science. For example, while OOD shifts in CV/NLP typically involve things like different color distributions or semantic contexts, in a PDE system, it is possible to get significantly different behavior of the PDE solution as physics parameters (and/or initial and boundary conditions) change. These can manifest as new features (for example, increased emphasis on very different frequency scales of the solution causing significant shifts in detail, sharpness, and other image attributes to be visible in the PDE solution) and depend on the physics coefficients in a highly non-linear fashion. One of our goals is to systematically quantify the TL performance as these OOD patterns emerge in our target solutions through modifying the physics parameters that control this data distribution. We visualize this diversity and sensitivity for our PDE systems in Appendix Fig A.1 to highlight the above point. In general, this diversity can be far greater in larger scientific systems (such as bifurcation in fluid flows where the dynamics can change dramatically with relatively small changes to the PDE inputs) and further amplified by ill-conditioned or numerically stiff systems. From a deep learning perspective, the dense prediction (and TL) of solutions exhibiting these characteristics with input numerical scales that can span orders of magnitude becomes a challenging task. One specific example of this that we mention in section 3 and appendix A.5 is our normalization strategy, which our results are quite sensitive to.
>
> ### R2-3: Interpreting error values and determining what level of accuracy is sufficient:
> This is an excellent question, and slightly difficult to answer as the level of acceptable error in most SciML applications is a very domain-specific consideration. For instance in some inverse problem applications, an error of O(1%) may be very good, especially for cases where the noise in data collection can be higher. But there are also problems in scientific computing such as fluid dynamics problems where many more digits of accuracy are needed to correctly model a phenomena and respect the problem boundaries/scales. In general the domain of PDEs is a data-limited scenario as running PDE solvers to generate data is expensive, so the cost-accuracy tradeoff (see e.g. de Hoop et al. 2022 https://arxiv.org/abs/2203.13181) is evaluated on a case-by-case basis (some PDEs are easier to solve than others). In other applications, data may come directly from observations and it is very expensive to collect these. The approach we take in the paper is to focus less on the absolute error values and more on benchmarking against the baseline of training from scratch. Interpreting results from this perspective is especially useful when considering the different training data requirements to reach a desired level of accuracy, as we show in Appendix Figure B.2. This figure helps demonstrate the dramatic advantages of pretraining, even when the downstream dataset is OOD, but we felt the simple l2 error used in most of our plots would be easier for most people to quickly interpret.

---

> > ### Comment · Reviewer_9JLh · 2023-08-16
> >
> > First of all, I would like to first thank the authors for providing the rebuttal.
> >
> > The authors have addressed my concerns appropriately, and I do not see significant issues from other reviews. I will suggest accepting this paper and will keep my original score.

---

### Official Review · Reviewer_Hq79 · 2023-07-28

**Soundness:** 3 good
**Presentation:** 3 good
**Contribution:** 2 fair
**Rating:** 6
**Confidence:** 2

**Summary:**

This paper explores the behavior and performance of neural operator models on multiple partial differential equation (PDE) systems in the transfer learning setting. The authors investigate the impact of various factors on the performance of neural operator models, including model size, downstream dataset size, underlying physics of the downstream tasks in relation to pre-training, and distributional shifts.

The study employs the "pre-train and fine-tune" paradigm for SciML (Scientific Machine Learning) problems. The authors show that it is possible and beneficial to develop more general SciML models capable of solving multiple tasks with the same set of weights, even when downstream tasks involve small-to-moderate distribution shifts relative to the pre-training data. The research demonstrates the potential of neural operator models for transfer learning, paving the way towards building foundation models for SciML.

**Strengths:**

Originality:

The paper demonstrates a high level of originality in several aspects. Firstly, it explores the use of neural operator models for solving multiple partial differential equation (PDE) systems, a novel approach that combines the power of neural networks with physics-informed learning. The authors investigate the behavior of these models under various conditions, including model size and downstream dataset size, providing valuable insights into their scalability and transferability. Secondly, the study applies the "pre-train and fine-tune" paradigm to SciML problems, showing the potential of developing more general SciML foundation models. This approach is innovative and opens up possibilities for efficiently addressing diverse downstream tasks with a single pre-trained model. Furthermore, the paper explores the impact of distributional shifts in downstream tasks relative to the pre-training data, contributing to the understanding of transfer learning behavior in the context of PDEs. The inclusion of multiple downstream PDE systems and the analysis of various performance metrics demonstrate a comprehensive and original investigation.

Clarity:

The paper is well-written and presents its findings in a clear and organized manner. The subsection is organized logically following the order of the posted questions.

Significance:

The significance of the paper lies in its contribution to the field of SciML and transfer learning for PDEs. By characterizing the behavior of neural operator models under different conditions and exploring transferability across multiple downstream PDE systems, the paper provides valuable insights for researchers working in the area of computational physics and scientific modeling. The findings have practical implications, as they suggest the potential for developing more versatile and efficient foundation models for SciML tasks. The "pre-train and fine-tune" approach demonstrated in the paper can be leveraged to achieve better generalization and performance in solving PDEs across a wide range of scenarios.

**Weaknesses:**

Motivation:

While the paper extensively analyzes the performance of neural operator models under different conditions, it lacks direct motivation of understanding the transfer learning for PDEs problem in SciML, as these different conditions have been well-explored in the LLM and CV community.

Potential Overfitting in Transfer Learning Settings:

The paper briefly mentions that few-shot transfer learning outperforms training from scratch, but it does not thoroughly address the possibility of overfitting to the specific downstream tasks during fine-tuning. An investigation into potential overfitting issues and strategies to mitigate them would be valuable.

**Questions:**

While exploring transferability across multiple downstream PDE systems is an intriguing and significant area of research, is there any differences in transfer learning between PDE problems and well-established domains like Language Model Learning (LLM) and Computer Vision (CV). What does the nature of PDEs and their associated physical systems introduce unique challenges and characteristic that may affect the transferability of learned representations and the generalization of neural operator models?

In my opinion, understanding and addressing these differences are crucial for insights from the transfer learning perspective in PDEs.

**Limitations:**

Listed in the Weaknesses and Limitations.

---

> ### Author Rebuttal · Authors · 2023-08-09
>
> We thank the reviewer for their thorough assessment and kind words regarding our work. The suggestions they provide are very helpful, and we address specific comments below:
>
> ### R1-1: Motivation for understanding the transfer learning (TL) for PDEs in SciML:
> We agree that the introduction can state our motivation more clearly. Given the success of the foundation model paradigm in the CV/NLP domains, a major motivation of ours is to provide an early systematic evaluation of TL and model/data scaling in the PDE domain. Solving PDE systems is a core component for many practical and applied problems such as climate modeling, molecular dynamics simulations, computational fluid dynamics, cancer growth modeling, etc. Fast and efficient solvers that can provide approximate solutions or serve as numerical preconditioners can be extremely valuable for these applications. Deep learning models have proven to be capable of serving this objective. However, the majority of current works train separate models for separate PDE systems. Here, TL and, specifically foundation models, offer an opportunity to reuse models and tailor them to specific downstream applications. This can be particularly useful in the PDE domain because the numerical solvers required to generate training data for many applications are very computationally expensive, leaving most efforts bottlenecked by limited data. Given the vast diversity of PDE systems and the range of scales/features in the solutions as well as coefficients of the PDEs, it is unrealistic to expect a single training set to span sufficiently diverse scenarios; in other words, we expect adapting models via fine-tuning will continue to be relevant for data-driven modeling of PDEs.  In the following section we expand on this point, as well as other important differences between TL in the PDE domain versus CV/NLP which we feel make our work especially important and timely for the SciML community.
>
> ### R1-2: Differences between transferability in PDEs vs. more traditional CV/NLP tasks:
> While we do mention some TL characteristics unique to PDEs in the text, we agree they could be emphasized more as a primary focus of the work. Other reviews also have similar suggestions. Broadly, a major difference in the PDE domain is the diversity of scales, features, and solution behaviors across PDE coefficients as well as systems of interest in science. For example, while OOD shifts in CV/NLP typically involve things like different color distributions or semantic contexts, in a PDE system, it is possible to get significantly different behavior of the PDE solution as physics parameters (and/or initial and boundary conditions) change. Capturing these is critical to any scientific objective. These can manifest as new features (for example, increased emphasis on very different frequency scales of the solution causing significant shifts in detail, sharpness, and other image attributes to be visible in the PDE solution) and depend on the physics coefficients in a highly non-linear fashion. One of our goals is to systematically quantify the TL performance as these OOD patterns emerge in our target solutions through modifying the physics parameters that control this data distribution. We visualize this diversity and sensitivity for our PDE systems in Appendix Fig A.1 to highlight the above point. In general, this diversity can be far greater in larger scientific systems (such as bifurcation in fluid flows where the dynamics can change dramatically with relatively small changes to the PDE inputs) and further amplified by ill-conditioned or numerically stiff systems. From a deep learning perspective, the dense prediction (and TL) of solutions exhibiting these characteristics with input numerical scales that can span orders of magnitude becomes a challenging task. One specific example of this that we mention in section 3 and appendix A.5 is our normalization strategy, which our results are quite sensitive to. We find that the normalization of physics coefficients, inputs, solutions to similar numerical ranges is very important and especially so while TL–choosing the pre-training dataset normalization for the fine-tuned model yields significantly better performance than the normalization from the downstream dataset/task.
>
> ### R1-3: Concerns about overfitting during few-shot fine-tuning:
> We agree it is critical to control for overfitting in the few-shot fine-tuning experiments, and that it should be called out more directly in the paper. Not surprisingly, for the larger model sizes we observe some overfitting in few-shot fine-tuning, especially on extreme OOD shifts, and we partially mitigate this by employing early stopping and tuning the learning rate for each specific model and fine-tuning dataset size. We broadly find that smaller learning rates work best for smaller downstream datasets. More generally, it is important to note that the fine-tuning strategy itself and how it may change with dataset size is an open question in this domain, and our work is intended as just a first step by systematically quantifying the TL behavior for a variety of representative PDE systems with the same fine-tuning procedure. It is very likely in the PDE domain there are alternate fine-tuning strategies which would work well for TL on OOD data, potentially involving supervision using physical laws directly from the PDE systems. As we mention briefly in Sec. 3, a notable alternative for the few-shot TL scenario is in-context learning, which is adopted in an interesting concurrent work (Yang et al. 2023, https://arxiv.org/pdf/2304.07993.pdf). While these directions are very interesting and warrant further research, we consider them out of scope for our investigation, and highlight them as limitations/topics for future work in Sec. 5.

---

> > ### Comment · Reviewer_Hq79 · 2023-08-20
> >
> > I would like to first thanks the authors for providing a detailed clarification for the motivation and difference. The authors have addressed my concerns appropriately, I decide to raise my score by 1 point.

---

### Author Rebuttal · Authors · 2023-08-10

We thank all reviewers for their insightful comments and suggestions. We attach additional figures for the model scaling experiments for Helmholtz and the additional data points for the extreme OOD test-case addressing R4-1 comments in Fig 1 and Fig 2, respectively. We refer the reviewers to the individual rebuttals for specific comments.

---

### Decision · Program_Chairs · 2023-09-21

**Decision:**

Accept (poster)

**Comment:**

This paper studies the transfer performance of the pre-trained foundation model for scientific machine learning applications described by PDEs, and it provides a comprehensive analysis of pre-training and transfer learning regimes over a diverse set of physical systems.

Most of the reviewers acknowledge the significance and novelty of the work, and find the presentation of the work clear and easy to follow.

In the final version, we suggest the authors carefully incorporate all the feedback provided by the reviewers.